# Wormhole Loss for Partial Shape Matching

**Amit Bracha**[*]  **Thomas Dagès**[*]  **Ron Kimmel**
Technion - Israel Institute of Technology
Haifa, Israel
`{amit.bracha,thomas.dages}@cs.technion.ac.il`

## Abstract

When matching parts of a surface to its whole, a fundamental question arises: Which points should be included in the matching process? The issue is intensified when using isometry to measure similarity, as it requires the validation of whether distances measured between pairs of surface points should influence the matching process. The approach we propose treats surfaces as manifolds equipped with geodesic distances, and addresses the partial shape matching challenge by introducing a novel criterion to meticulously search for consistent distances between pairs of points. The new criterion explores the relation between intrinsic geodesic distances between the points, geodesic distances between the points and surface boundaries, and extrinsic distances between boundary points measured in the embedding space. It is shown to be less restrictive compared to previous measures and achieves state-of-the-art results when used as a loss function in training networks for partial shape matching.

## 1   Introduction

Shape correspondence is a core challenge in computer graphics and computer vision, distinguished by its wide array of applications extending from 3D modeling and animation to object recognition and beyond. The task involves establishing mappings between corresponding points across different surfaces that undergo non-rigid transformations. Shape correspondence grows particularly intricate when it involves partial surfaces, where the objective is to find correspondences between parts of surfaces or between a complete surface and a part of another. This sub-task is particularly difficult as it involves incomplete data and possible different topology of the matched surfaces. Consequently, training neural networks for the partial matching task is challenging, especially for unsupervised methods, as there is no information about which parts are missing and which should be matched.

The state of the art in partial shape matching is performed with Functional Maps (FM), a popular framework for relating functions defined on surfaces via basis functions defined on the surfaces [2, 18, 22, 25, 36, 45, 34]. Recently, it was argued that the unused information from the full surface impairs the learning process when FM is incorporated in the learning pipeline [10], with better results obtained when bypassing FM altogether by direct estimation of correspondences. This method is primarily guided by a loss function preserving the pairwise geodesic distances [25], which relies on the fact that such distances are preserved under isometric transformations. When dealing with partial surfaces, not all distances are preserved, as minimal geodesics on the full surface may go through a region which is missing in the partial surface. Therefore, undesired biases are integrated into the learning procedure, which can impair the quality of the learned correspondence. This applies to both FM-based unsupervised loss functions [48] and geodesic-based ones [25], where isometry between the surfaces is assumed.

---

[*]Equal contribution

38th Conference on Neural Information Processing Systems (NeurIPS 2024).

In this paper, we present a novel loss for partial shape correspondence that takes into account partial surfaces. It is constructed upon the notion of *consistent* pairs of points, where a pair is said to be consistent if the geodesic distance between the points is the same on the partial and full surface. Our task is to define a criterion that captures as much as possible consistent pairs, where these pairs are referred to as *guaranteed* pairs. Using such a criterion, we can limit our matching procedures to consider only guaranteed pairs, thereby avoiding distortions. Filtering out potentially inconsistent distances between pairs of points was first studied in [13, 12, 14, 46, 47]. In particular, in [12, 46] the authors showed that for non-Euclidean spaces the geodesic distance between two points on a manifold is preserved when this distance is smaller than the sum of the surface distances of each point to the boundary of the surface. We show that this criterion is too conservative and it filters out a substantial number of consistent pairs of points. It can be improved by using the extrinsic information encapsulated in the manifold's embedding space. The new condition can be shown to provide significantly more guaranteed pairs of points.

Using this novel criterion to find consistent pairs of points, we present a new loss function for unsupervised shape correspondence tailored for partial surfaces. It is based on the ratio between the geodesic pairwise distance, that is calculated on the partial surface, and a measure defined by our criterion, which involves distances to points on the boundaries and the distance in the embedding space between the corresponding boundary points. The new loss was used for training an unsupervised shape correspondence neural network achieving state-of-the-art (SOTA) results on the reference SHREC'16 CUTS and HOLES datasets [20] and on the recent PFAUST dataset [10]. Our code can be found at https://github.com/ABracha/Wormhole.

**Contributions**

- We introduce a novel criterion for identifying consistent pairs, which are pairs of points between which the geodesic distance is the same for the full and the partial surfaces.
- Using this criterion, we create a novel unsupervised loss function tailored specifically for partial surfaces, achieving SOTA results on partial shape correspondence benchmarks.

## 2  Related Efforts

Early attempts to solve shape correspondence involved extracting hand-crafted features for each point [4, 55]. Leveraging the intrinsic nature of the Laplace-Beltrami operator (LBO), its eigenfunctions were often employed to define such features on the manifold, invariant to non-rigid isometry, thereby, removing the dependency on the manifold's embedding in Euclidean space. Later, the Functional-Map (FM) framework was proposed [40]. It builds on the fact that the mapping of consistent feature representations in local bases is linear, even for complex non-rigid spatial deformations of the manifolds. In fact, the first neural network for solving shape correspondence was based on the FM framework [33], which introduced a differentiable FM-layer to learn the correspondence map.

More recent methods suggested two alternative approaches for unsupervised shape matching. The first is based on the fact that isometries preserve geodesic distances [25], and the second is based on properties satisfied by the FM operator for isometric transformations [48]. Improved architectures followed [22, 53, 31, 54]. Alternative loss functions were suggested, such as a supervised contrastive loss [30], or using the objective function from [2], which penalizes the difference between the FM output from the FM-layer and the FM estimated from the point-to-point mapping [18]. Other papers explored different metrics, such as the scale invariant metric [11, 41], or anisotropic Riemannian metrics using the Finsler-based LBO [61].

Several efforts were made to tackle surface matching under partiality. Some approaches are based on a search strategy to find either the missing parts [7, 6, 43], or direct correspondence by exhaustive search [24]. Few methods learn to find the correspondence, one such example learned linear invariant bases suited for partial surfaces rather than the classical LBO eigenspaces [36]. Another supervised learning approach applied an attention layer [59] on extracted features, allowing feature interaction between the full and partial surfaces prior to the FM-layer [3]. An unsupervised learning method, based on preserving intrinsic distances, avoided the use of an FM-layer altogether [10]. The motivation was that FM introduces noise when used to match the whole to a part of a surface.

When dealing with partial surfaces, one needs to consider the fact that distances can be significantly different when measured between corresponding points in the partial and the whole surface. Thus,

encouraging distance preservation between corresponding points can introduce undesired biases on the learned matching. Although [10] reported the current SOTA, it only tackles part of the problem. Here, we introduce a novel loss function based on *consistent* pairs of points and geodesic distances, greatly reducing the effect of inconsistent distances on the learned correspondence.

To construct our loss function, we first revisit the notion of consistent pairs. The link between geodesic distances and distances to the boundaries of flat and non-flat surfaces was studied in [12, 14, 46, 47]. Validity conditions were suggested to predict whether distances between matching points are consistent between the partial and the full surface. Here, we revisit and refine these conditions, and introduce a more inclusive criterion that guarantees more consistent pairs by which the matching results improve.

## 3 On the Consistency of Distances between Pairs of Points in Partial Surfaces

### 3.1 Consistent and Guaranteed Pairs of Points

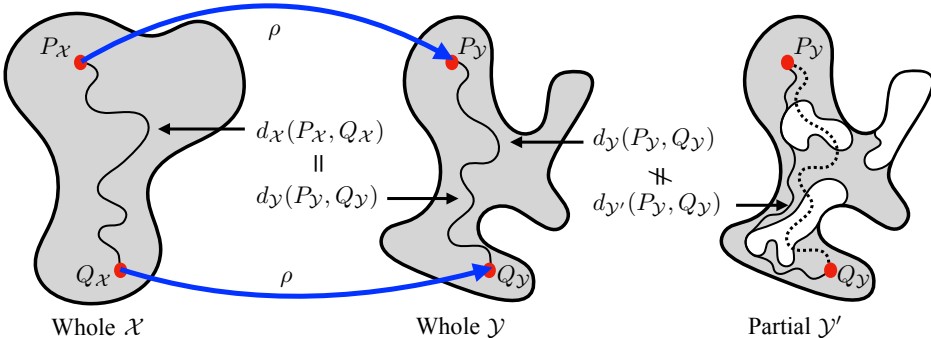

Figure 1: For a distance preserving map between full surfaces $\mathcal{X}$ and $\mathcal{Y}$, also known as an isometry, the minimal geodesics in the partial version $\mathcal{Y}'$ may not correspond to those in the full surfaces $\mathcal{Y}$. That is, the geodesic distances between corresponding points may get larger.

In this section, we present the concept of *consistent* and *guaranteed* pairs of points.

Let $\mathcal{X}$ and $\mathcal{Y}$ be isometric surfaces with a bijective mapping $\rho : \mathcal{X} \to \mathcal{Y}$. In partial shape matching, the surfaces may have missing parts, meaning that we only have access to $\mathcal{X}' \subset \mathcal{X}$ and $\mathcal{Y}' \subset \mathcal{Y}$. For simplicity, we assume that we have access to the full surface $\mathcal{X}$ but not the full surface $\mathcal{Y}$ and so $\mathcal{X}' = \mathcal{X}$ and $\mathcal{Y}' \neq \mathcal{Y}$. We can easily generalise the discussion if $\mathcal{X}' \neq \mathcal{X}$. Consider two points $P_{\mathcal{Y}}$ and $Q_{\mathcal{Y}}$ on the partial surface $\mathcal{Y}'$ and their corresponding points $P_{\mathcal{X}} = \rho^{-1}(P_{\mathcal{Y}})$ and $Q_{\mathcal{X}} = \rho^{-1}(Q_{\mathcal{Y}})$ on $\mathcal{X}$. As the mapping $\rho$ is isometric, the geodesic distances, reflecting the length of the shortest paths, on the full surfaces $d_{\mathcal{X}}$ and $d_{\mathcal{Y}}$ are equal by definition, $d_{\mathcal{X}}(P_{\mathcal{X}}, Q_{\mathcal{X}}) = d_{\mathcal{Y}}(P_{\mathcal{Y}}, Q_{\mathcal{Y}})$. However, it may be that the shortest path on the full surface $\mathcal{Y}$ between $P_{\mathcal{Y}}$ and $Q_{\mathcal{Y}}$ passes through a missing part of $\mathcal{Y}'$. In such a case, computing the geodesic path between these two points could lead to a longer path than the one on the full surface, implying that $d_{\mathcal{Y}}(P_{\mathcal{Y}}, Q_{\mathcal{Y}}) \neq d_{\mathcal{Y}'}(P_{\mathcal{Y}}, Q_{\mathcal{Y}})$ and that, in turn, $d_{\mathcal{X}}(P_{\mathcal{X}}, Q_{\mathcal{X}}) \neq d_{\mathcal{Y}'}(P_{\mathcal{Y}}, Q_{\mathcal{Y}})$, even if the original mapping $\rho$ is isometric (see Figure 1). As such, methods relying on or enforcing the preservation of geodesic distances based on the isometry of the transformation $\rho$ are inappropriate when dealing with partial surfaces. They could incorrectly enforce equal distances when they should not. Although not all geodesic distances are preserved for partial surfaces, many nevertheless are preserved, leading to the definition of *consistent* pairs.

**Definition 1** (Consistent Pair of Points). A pair of points $P_{\mathcal{Y}}$ and $Q_{\mathcal{Y}}$ of a partial surface $\mathcal{Y}'$ is said to be *consistent* if the geodesic distance on the partial surface is the same as that on the original full surface, i.e. $d_{\mathcal{Y}}(P_{\mathcal{Y}}, Q_{\mathcal{Y}}) = d_{\mathcal{Y}'}(P_{\mathcal{Y}}, Q_{\mathcal{Y}})$.

As there are many consistent pairs even in cases of cuts and partiality, methods enforcing distance preservation between all pairs, even inconsistent ones, perform fairly well. However, penalizing distance dissimilarity for inconsistent pairs is undesired and decreases the quality of the resulting mapping. Our goal is to filter the set of pairs by finding as many as possible consistent pairs, and then relying only on these *guaranteed* pairs when matching surfaces.

**Definition 2** (Guaranteed Pair of Points). A pair of points $P_{\mathcal{Y}}$ and $Q_{\mathcal{Y}}$ of a partial surface $\mathcal{Y}'$ is said to be *guaranteed* with respect to a criterion $\mathcal{C} : \mathcal{Y}' \times \mathcal{Y}' \to \{0, 1\}$, or $\mathcal{C}$-guaranteed in short, if the criterion proves that the pair is consistent.

Note, that the criterion is unsupervised as no oracle provides knowledge from the full surface $\mathcal{Y}$ to compute it. The better the criterion, the more consistent pairs it finds, allowing for more consistent information to be used for finding the matching, see Figure 2. In the rest of this section, we focus on the search for consistent pairs in the partial surface $\mathcal{Y}'$ as a preprocessing step and will return to the surface $\mathcal{X}$ when we perform partial shape matching.

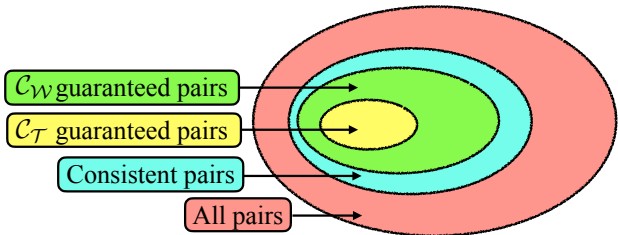

Figure 2: Venn diagrams showing the relation between all pairs of points, consistent and guaranteed pairs for a surface with boundaries. All guaranteed pairs are consistent. Our criterion $\mathcal{C}_{\mathcal{W}}$ is more inclusive than that of [12, 46], $\mathcal{C}_{\mathcal{T}}$. All guaranteed pairs by $\mathcal{C}_{\mathcal{T}}$ are also guaranteed by $\mathcal{C}_{\mathcal{W}}$.

### 3.2 Expanding the Set of Guaranteed Pairs via Extrinsic Distances between Boundary Points

We introduce a criterion that finds as many consistent pairs as possible, which we then denote as guaranteed. Denote $\mathcal{B} \subset \mathcal{Y}'$ the boundary of $\mathcal{Y}'$. For simplicity, we treat all boundaries equally. In [12, 46], the criterion, $\mathcal{C}_{\mathcal{T}}$, was designed to reject pairs of points, $P_{\mathcal{Y}}$ and $Q_{\mathcal{Y}}$ on $\mathcal{Y}'$, for which the sum of their distances on to the boundary $\mathcal{B}$ is less than the geodesic distance between them,

$$\mathcal{C}_{\mathcal{T}}(P_{\mathcal{Y}}, Q_{\mathcal{Y}}) = \begin{cases} 1 & \text{if } d_{\mathcal{Y}'}(P_{\mathcal{Y}}, Q_{\mathcal{Y}}) \leq d_{\mathcal{Y}'}(P_{\mathcal{Y}}, \mathcal{B}) + d_{\mathcal{Y}'}(Q_{\mathcal{Y}}, \mathcal{B}) \\ 0 & \text{otherwise,} \end{cases} \tag{1}$$

where $d_{\mathcal{Y}'}(P_{\mathcal{Y}}, \mathcal{B})$ is the minimal length of the minimal geodesic of $P_{\mathcal{Y}}$ to any boundary point,

$$d_{\mathcal{Y}'}(P_{\mathcal{Y}}, \mathcal{B}) = \min_{B \in \mathcal{B}} d_{\mathcal{Y}'}(P_{\mathcal{Y}}, B). \tag{2}$$

This condition naturally follows from the fact that the length of a path between the points in the full surface $\mathcal{Y}$ that passes through the boundary $\mathcal{B}$ is at least as long as the sum of their distances to the boundary.

Note, that this criterion is overly conservative and removes many consistent pairs. It practically ignores the length of possible trajectories connecting the boundary points, see Figure 3. To mitigate this condition, we propose to include extrinsic information that bounds from below the distances between pairs of boundary points. Note that when solving intrinsic problems, the practice of combining intrinsic and extrinsic information is uncommon, yet it has been shown to be beneficial for other types of robustness [16]. For simplicity, we assume from now on that the manifold is embedded in a Euclidean space. We then readily use the following straightforward relation between geodesic distances and Euclidean ones.

**Theorem 1** (Euclidean bound). *The geodesic distance between any points $P$ and $Q$ on a surface $\mathcal{Y} \in E$, for some embedding Euclidean space $E = \mathbb{R}^n$, is larger than or equal to the Euclidean distance between the points measured in the embedding space,*

$$d_{\mathcal{Y}}(P, Q) \geq d_E(P, Q),$$

*where $d_E(P, Q) = \|P - Q\|_2$.*

For a proof see Appendix A.1. We propose to consider trajectories passing through any pairs of boundary points $B_1$ and $B_2$ in $\mathcal{B}$. By acting like a *wormhole* connecting two boundary points, the length of the straight line in $E$ connecting $B_1$ and $B_2$ can serve as a lower bound of the geodesic distance on the full surface between the points. Geodesic distance measures the distance of $P$ and $Q$

to boundary points $B_1$ and $B_2$, whereas the Euclidean distance $d_E(B_1, B_2)$ bounds from below the geodesic distance on the full surface between $B_1$ and $B_2$. Our *wormhole* criterion $\mathcal{C}_\mathcal{W}$ is given by

$$\mathcal{C}_\mathcal{W}(P_\mathcal{Y}, Q_\mathcal{Y}) = \begin{cases} 1 & \text{if } d_{\mathcal{Y}'}(P_\mathcal{Y}, Q_\mathcal{Y}) \leq \min_{B_1, B_2 \in \mathcal{B}} d_{\mathcal{Y}'}(P_\mathcal{Y}, B_1) + d_{\mathcal{Y}'}(Q_\mathcal{Y}, B_2) + d_E(B_1, B_2), \\ 0 & \text{otherwise.} \end{cases}$$

(3)

By design, this criterion can be used to find consistent pairs.

**Theorem 2** ($\mathcal{C}_\mathcal{W}$ guarantees). *The wormhole criterion $\mathcal{C}_\mathcal{W}$ yields guaranteed pairs.*

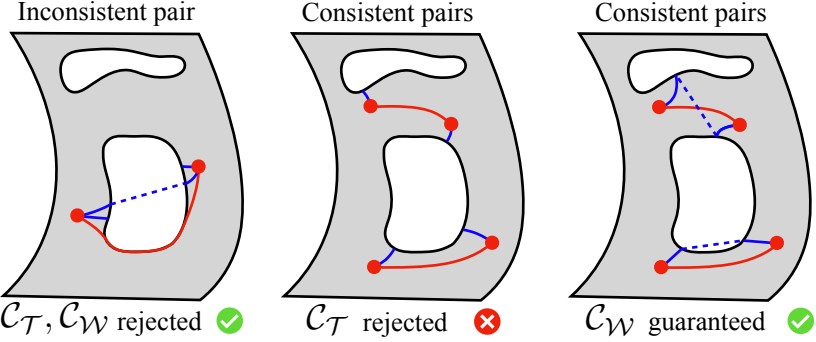

Figure 3: Example of inconsistent and consistent pairs of points. The minimal geodesic paths are colored red, while paths to the boundary points are colored blue. Different boundary points are selected in $\mathcal{C}_\mathcal{T}$ [12, 46] and $\mathcal{C}_\mathcal{W}$. The Euclidean lines connecting the boundary points selected by $\mathcal{C}_\mathcal{W}$ are dashed blue. Both criteria correctly reject inconsistent pairs (left). Since $\mathcal{C}_\mathcal{T}$ ignores the distance between boundary points, it discards many consistent pairs (middle). Criterion $\mathcal{C}_\mathcal{W}$ finds more consistent pairs by including the extrinsic Euclidean distance between boundary points (right).

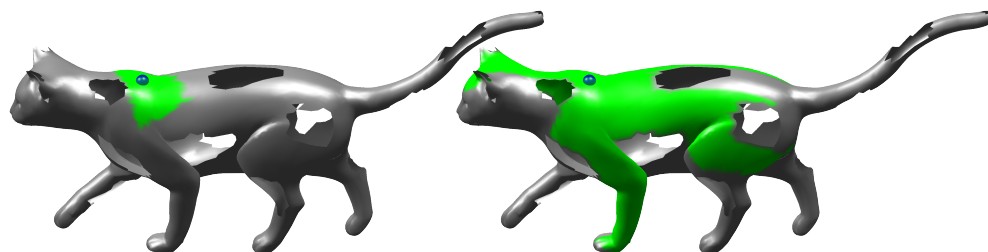

Figure 4: We plot in green the set of points that together with the blue point satisfy $\mathcal{C}_\mathcal{T}$ [12, 46] (left) and $\mathcal{C}_\mathcal{W}$ (right). The blue point and any of the green ones form a guaranteed pairs. The cat surface is taken from the SHREC'16 HOLES dataset [20].

See Appendix A.2 for a proof. The new criterion for finding consistent pairs, generalizes the best known criterion $\mathcal{C}_\mathcal{T}$ as all $\mathcal{C}_\mathcal{T}$-guaranteed pairs are also $\mathcal{C}_\mathcal{W}$-guaranteed, see, e.g. Figure 4.

So far, we implicitly assumed that the metric on the manifold is the one induced from the Euclidean metric of the embedding space, meaning that curves have the same length from the perspective of the manifold and the embedding space. We can generalize the wormhole criterion to handle general Riemannian metrics on the manifold, by modulating the Euclidean distance of the straight line between boundary points by the minimal scaling relation between arclength measured by the embedding metric restricted to the manifold and one measured by the Riemannian metric of the embedded manifold. We present our generalised metric-sensitive criterion in Appendix A.3

The definition of consistent pairs has been in a continuous setting. We need to adapt it to the discrete world. We describe in Appendix B.2 how to do so. In short, we only consider points on the surface or its boundary to belong to a finite sampled set $V$ of vertices on the continuous surface. We construct the threshold and binary mask matrices $\boldsymbol{K}$ and $\boldsymbol{M}$ of size $|V| \times |V|$, containing at entry $ij$ respectively the threshold of the criterion $\mathcal{C}_\mathcal{W}$ and the binary criterion value $\mathcal{C}_\mathcal{W}$ of the $ij$ pair of vertices,

$$\boldsymbol{K}_{ij} = \min_{B_1, B_2 \in \mathcal{B}} d_{\mathcal{Y}'}(v_i, B_1) + d_{\mathcal{Y}'}(v_j, B_2) + d_E(B_1, B_2)$$

(4)

and

$$\boldsymbol{M}_{ij} = \mathbb{1}_{(\boldsymbol{D}_{\mathcal{Y}'})_{ij} \leq \boldsymbol{K}_{ij}}, \tag{5}$$

where $\mathbb{1}$ is the indicator function and $(\boldsymbol{D}_{\mathcal{Y}'})_{ij}$ is the computed geodesic distance on the partial surface $\mathcal{Y}'$ between $v_i$ and $v_j$.

To evaluate the quality of our criterion $\mathcal{C}_{\mathcal{W}}$, we searched empirically for consistent pairs using either our criterion or $\mathcal{C}_{\mathcal{T}}$ on the PFAUST-M and PFAUST-H [10] datasets, the latter comprising of shapes with more holes than the former. We provide in Table 1 the percentage of consistent pairs, and among those the percentage of guaranteed pairs by the different criteria, along with the standard deviations. Our criterion $\mathcal{C}_{\mathcal{W}}$ is able to recover most pairs and it guarantees twice as many as the previous criterion $\mathcal{C}_{\mathcal{T}}$.

Table 1: Empirical quantitative evaluation of the ability to recover consistent pairs. The numbers represent either the average percentage of consistent pairs out of the total number of pairs of points in a discrete shape, or the average percentage of guaranteed pairs by criteria $\mathcal{C}_{\mathcal{T}}$ and $\mathcal{C}_{\mathcal{W}}$ out of the number of consistent pairs. In parenthesis we provide the standard deviation across different shapes. Averages and standard deviations are computed within the PFAUST-M or PFAUST-H [10] datasets, the latter possessing shapes with more holes.

| Dataset | %Consistent | %Guaranteed ($\mathcal{C}_{\mathcal{T}}$) [12, 46] | %Guaranteed ($\mathcal{C}_{\mathcal{W}}$) (Ours) |
|---------|-------------|------------------------|------------------------|
| PFAUST-M | 78 ($\pm 16$) | 48 ($\pm 18$) | **82** ($\pm 14$) |
| PFAUST-H | 53 ($\pm 16$) | 30 ($\pm 18$) | **65** ($\pm 18$) |

## 4 Applications

### 4.1 Multi-Dimensional Scaling

In this paper, we concentrate on partial shape matching. Additionally, we demonstrate that the proposed criterion for identifying consistent pairs can be beneficial for other tasks. Specifically, we show how it can be incorporated into a multidimensional scaling (MDS) pipeline. The MDS goal is to embed curved manifolds into a low dimensional Euclidean space $\mathbb{R}^m$, for some small $m$, such that the pairwise Euclidean distances of the new embedding are as close as possible to the original geodesic distances between corresponding pairs of points. To obtain embeddings robust to missing parts, we handle boundary conditions on partial surfaces by relying on consistent distances of guaranteed pairs. In [46, 47], the TCIE method minimises a masked distance preservation objective, with mask weights $w_{ij} = \mathcal{C}_{\mathcal{T}}(P_i, P_j)$. Our MDS method, named *wormhole constrained isometric embedding* (WHCIE), adapts this approach by replacing the mask weights with our criterion $w_{ij} = \mathcal{C}_{\mathcal{W}}(P_i, P_j) = \boldsymbol{M}_{ij}$.

We plot the resulting WHCIE planar embedding of toy surfaces in Figure 5. For each Swiss roll, we removed either a hole or a full rectangular cut connected to the boundary. We compare with several reference manifold learning techniques: Isomap [57], MLLE [62], Laplacian eigenmaps [5], UMAP [37], and TCIE [46, 47]. Methods that do not check for distance consistency between pairs of points yield distorted embeddings. The wormhole criterion finds significantly more consistent pairs compared to the TCIE, resulting in an embedding similar to that of the actual surface. That is, a flat rectangle with a hole or a cut. For more details and comparison with related methods, see Appendix B.1.

### 4.2 Partial Shape Matching

We next return to our primary application - matching shapes. To that end, we introduce the *wormhole loss,* a method based on guaranteed pairs specifically designed for partial shape matching.

#### 4.2.1 Wormhole Loss: Unsupervised Partial Shape Matching using Consistent Pairs

Our mask $\boldsymbol{M}$ and threshold $\boldsymbol{K}$ matrices operate on minimal geodesic distances. A relevant unsupervised loss function using such distances for near isometric surfaces $\mathcal{X}$ and $\mathcal{Y}$ is [25],

$$\mathcal{L}_{\text{geo}}(\boldsymbol{P}, \boldsymbol{D}_{\mathcal{X}}, \boldsymbol{D}_{\mathcal{Y}}) = \frac{1}{|\mathcal{Y}|^2} \left\| \boldsymbol{P} \boldsymbol{D}_{\mathcal{X}} \boldsymbol{P}^{\top} - \boldsymbol{D}_{\mathcal{Y}} \right\|_F^2, \tag{6}$$

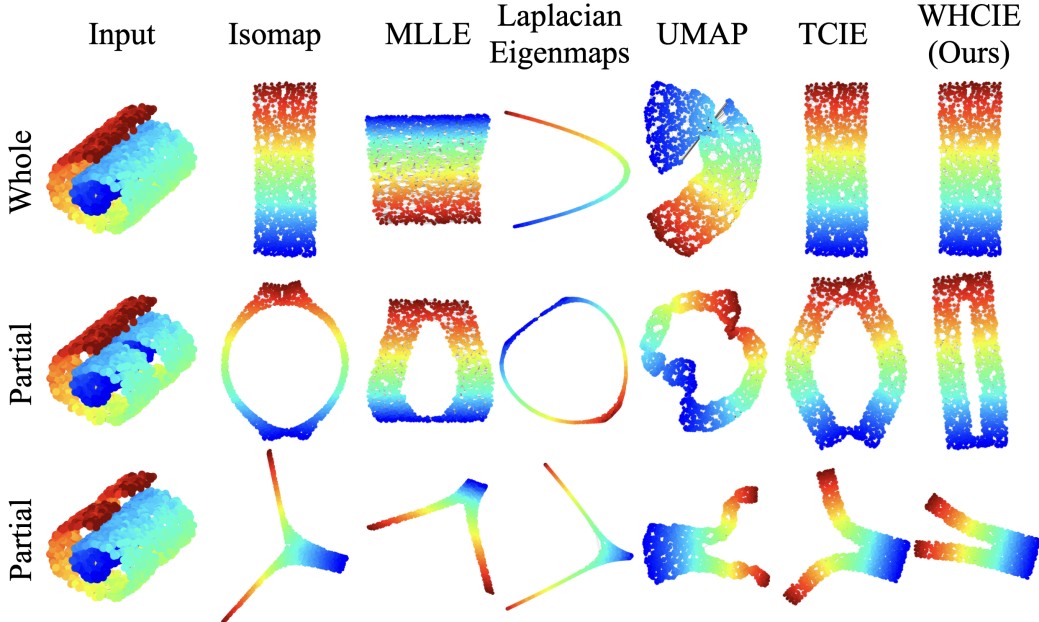

Figure 5: MDS embedding of various manifold learning methods on a whole Swiss roll (top) and on two versions of it with Gaussian noise, having either a rectangular hole (middle) or a rectangular cut (bottom). Other methods are referred to in Figures 7 and 8 in Appendix B.1.

where $\|.\|_F$ is the Frobenius norm, $\boldsymbol{P}$ is the estimated point-to-point soft correspondence matrix, $\boldsymbol{D}_{\mathcal{X}}$ is the geodesic distance matrix on surface $\mathcal{X}$, and $|\mathcal{Y}|$ is the area of surface $\mathcal{Y}$. The stochastic matrix $\boldsymbol{P}$ of size $|V_{\mathcal{Y}}| \times |V_{\mathcal{X}}|$ takes values in $\boldsymbol{P}_{ij} \in [0, 1]$, and is row-normalized such that $\sum_i \boldsymbol{P}_{ij} = 1$. This continuous setting accounts for possible different samplings of the surfaces, as the matched part in $\mathcal{X}$ of each sampled vertex in $V_{\mathcal{Y}}$ may not have a corresponding sample point in $V_{\mathcal{X}}$. If $\boldsymbol{P}$ is estimated to be the correct correspondence matrix, then, the distances are preserved and the loss is minimal. We incorporate our mask $\boldsymbol{M}$ into this loss function for partial surfaces $\mathcal{Y}'$ as follows,

$$\mathcal{L}_{\text{geo}}(\boldsymbol{P}, \boldsymbol{D}_{\mathcal{X}}, \boldsymbol{D}_{\mathcal{Y}'}, \boldsymbol{M}) = \sum_{ij} \boldsymbol{M}_{ij} \boldsymbol{A}_{\mathcal{Y}'ii} \boldsymbol{A}_{\mathcal{Y}'jj} (\boldsymbol{P}\boldsymbol{D}_{\mathcal{X}}\boldsymbol{P}^{\top} - \boldsymbol{D}_{\mathcal{Y}'})^2_{ij}, \tag{7}$$

where $\boldsymbol{P}$, now a matrix of size $|V_{\mathcal{X}}| \times |V_{\mathcal{Y}'}|$ is a full-to-partial correspondence matrix, and $\boldsymbol{A}_{\mathcal{Y}'}$ is diagonal matrix of the vertex areas of shape $\mathcal{Y}'$. Thus, $\boldsymbol{P}\boldsymbol{D}_{\mathcal{X}}\boldsymbol{P}^{\top}$ of size $|V_{\mathcal{Y}'}| \times |V_{\mathcal{Y}'}|$ maps the distances between pairs of vertices in $\mathcal{X}$ to distances between pairs of vertices in $\mathcal{Y}'$. The mask matrix $\boldsymbol{M}$ filters out inconsistent pairs, allowing the loss to be minimized with distances transferred from the full surface matched with those on the partial surface. This loss generalises the one proposed in [10] by incorporating masking weights. We provide full details on how to derive this loss in the supplementary material Appendix B.3.

In our experiments, we found that the binary criterion can be relaxed to better use the interaction between pairs. Many inconsistent pairs have distances only slightly increased by the presence of holes, thus, filtering them out removes valuable, though slightly noisy, information. We propose to regularize the binary mask matrix into a soft mask $\boldsymbol{M}^s$ as follows,

$$\boldsymbol{M}^s_{ij} = \min\left(\frac{\boldsymbol{K}_{ij}}{(\boldsymbol{D}_{\mathcal{Y}'})_{ij}}, 1\right). \tag{8}$$

The weights of the soft mask matrix inhibit the contribution to the loss of non-guaranteed pairs whose distance is significantly larger than the theoretical threshold of our criterion $\mathcal{C}_{\mathcal{W}}$. By the same token, it preserves the influence of distances of non-guaranteed pairs which are near the threshold.

### 4.2.2 Implementation Considerations

We evaluate the new geometric-wormhole loss, $\mathcal{L}_{\text{geo}}$, by integrating it into a pipeline adapted from [10], replacing their primary loss component, first introduced in [25]. Our pipeline takes raw input

features and then refines them independently for the surfaces using a shared-weights neural network, DiffusionNet [54]. These refined features are then matched using Softmax similarity [7, 18] to directly compute a correspondence matrix $P$, which is then plugged into the geometric-wormhole loss. We keep the regularization loss [48] promoting the preservation of vertex area between corresponding vertices, which is computed with the functional map extracted from the correspondence matrix. We provide further details of the proposed pipeline and implementation details in the supplementary material Appendix B.4.

### 4.2.3 Evaluation

Table 2: Quantitative analysis on SHREC'16 [20]. The numbers represent the average geodesic error (multiplied by 100) of the results following post-processing refinement. The best performance is highlighted in bold, and the second best is underlined. These results underscore the proposed method's dominance in unsupervised shape matching. It surpasses both supervised, unsupervised, and pretrained methods on the HOLES benchmark, known for its highly challenging scenarii.

| Test-set | CUTS | | HOLES | |
|---|---|---|---|---|
| Training-set | CUTS | HOLES | CUTS | HOLES |
| Axiomatic methods | | | | |
| PFM [45]→Zoomout | $9.7 \rightarrow 9.0$ | | $23.2 \rightarrow 22.4$ | |
| FSP [34] → Zoomout | $16.1 \rightarrow 15.2$ | | $33.7 \rightarrow 32.7$ | |
| Supervised methods | | | | |
| GeomFMaps [22] → Zoomout | $12.8 \rightarrow 10.4$ | $19.8 \rightarrow 16.7$ | $20.6 \rightarrow 17.4$ | $15.3 \rightarrow 13.0$ |
| DPFM [3] → Zoomout | $3.2 \rightarrow 1.8$ | $8.6 \rightarrow 7.4$ | $15.8 \rightarrow 13.9$ | $13.1 \rightarrow 11.9$ |
| Pretrained using external datasets | | | | |
| RobustFMnet with Refinement [18] | 3.2 | 5.6 | 13.5 | 8.2 |
| Unsupervised methods | | | | |
| Unsupervised-DPFM [3] → Zoomout | $11.8 \rightarrow 12.8$ | $19.5 \rightarrow 18.7$ | $19.1 \rightarrow 18.3$ | $17.5 \rightarrow 16.2$ |
| RobustFMnet [18] → Refinement | $16.9 \rightarrow 10.6$ | $22.7 \rightarrow 16.6$ | $18.7 \rightarrow 16.2$ | $23.5 \rightarrow 18.8$ |
| DirectMatchNet [10] → Refinement | $\textbf{6.9} \rightarrow 5.6$ | $12.2 \rightarrow 8.0$ | $\textbf{14.2} \rightarrow \textbf{10.2}$ | $\underline{11.4} \rightarrow \underline{7.9}$ |
| DirectMatchNet LPF [10] → Refinement | $7.1 \rightarrow \underline{4.7}$ | $\textbf{8.6} \rightarrow \textbf{5.5}$ | $16.4 \rightarrow 11.6$ | $12.3 \rightarrow 8.6$ |
| Wormhole (Ours) → Refinement | $\textbf{6.9} \rightarrow \textbf{4.3}$ | $\underline{10.8} \rightarrow \underline{7.2}$ | $17.1 \rightarrow \underline{10.9}$ | $10.9 \rightarrow \textbf{6.6}$ |

**Datasets.** We evaluate our method on the benchmarks SHREC'16 [20] and PFAUST [10]. SHREC'16 includes two datasets, CUTS and HOLES. Both datasets contain processed figures from the TOSCA [15] dataset. In CUTS, the figures were cut using several 3D planes. In HOLES, besides the cuts made by the planes, additional holes were added, resulting in partial surfaces with more topological changes and a longer boundary, which makes this dataset particularly harder for shape correspondence. The second benchmark on which we evaluate our method is PFAUST, recently created by [10]. It contains figures from the FAUST-Remeshed dataset [44] that were processed by creating holes in them. This benchmark is also divided into two datasets; in PFAUST-M, there are bigger but fewer holes, and in the PFAUST-H, there are smaller but more numerous holes. The latter presents a harder task for shape correspondence, as a greater number of holes results in more significant changes in the topology.

**Baselines.** The baseline methods we evaluated fall into three categories: Axiomatic methods – PFM [45] and FSP [34], supervised methods – GeomFMaps [22] and DPFM [3], which is the current SOTA for the CUTS and PFAUST datasets, and unsupervised methods – unsupervised DPFM [3], RobustFmaps [18], and DirectMatchNet [10], which is the current SOTA for the HOLES and the unsupervised SOTA on the PFAUST datasets. We had to re-implement the loss function for unsupervised DPFM due to source code unavailability. Recently, the authors of RobustFmaps updated their models to include a pretraining phase including four external datasets. For completeness, we include this pretrained version in our results table, albeit we separate it from the other methods not using any external datasets. For fairness, we also provide the results of RobustFmaps when trained only on HOLES or CUTS without any pretraining, following the common protocol of the benchmarks. For post-processing refinement, we employed either Zoom-out [38] or test time adaptation refinement [18] depending on what the original papers used.

**Results.** The quantitative results shown in Tables 2 and 3 indicate that our method reaches state-of-the-art performance for unsupervised partial shape correspondence. We also display superior qualitative performance in Figure 6 and Appendix B.5, demonstrating our method's robustness to missing parts. Indeed, unlike other methods our mappings are hardly distorted at challenging locations, such as points close to boundaries. A major improvement over other approaches was in the more challenging datasets HOLES, PFAUST-H, and PFAUST-M, whereas in the CUTS dataset a relatively modest improvement is shown. The missing parts in the CUTS dataset were created by slicing figures with 3D planes. In practice, this procedure does not modify much the topology of the surface and it does not introduce many inconsistent pairs. As such, cuts mostly preserve all geodesic distances, thus, our novel loss function was almost identical to the one in Equation (6) [25]. On the more challenging datasets, a larger part of the inter-geodesic distances differ between the partial and full surfaces, enabling our method to demonstrate its strength, even in comparison to supervised methods or pretrained models on external datasets. In fact, our unsupervised approach demonstrates superior performance to the best supervised approaches on the HOLES and PFAUST-H benchmarks.

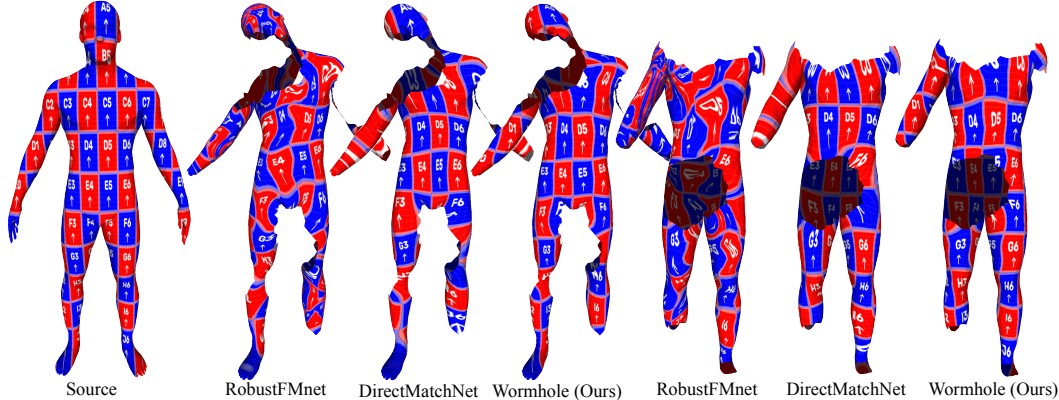

Source RobustFMnet DirectMatchNet Wormhole (Ours) RobustFMnet DirectMatchNet Wormhole (Ours)

Figure 6: Qualitative results on the test set of PFAUST-H. Our method yields less distortions near boundaries demonstrating its robustness to missing parts.

Table 3: Quantitative results on PFAUST [10]. The numbers indicate average geodesic error ($\times 100$). Our method surpasses previous unsupervised shape correspondence methods on the medium and hard datasets, and is on par with the supervised method on the hard dataset. The largest improvement is on the hard dataset, showing robustness to topological changes due to missing parts of the surface.

|              |                        | PFAUST-M | PFAUST-H |
|--------------|------------------------|----------|----------|
| Supervised   | DPFM [3]               | 3.0      | 6.8      |
| Unsupervised | Unsupervised-DPFM [3]  | 9.3      | 12.7     |
|              | RobustFMnet [18]       | 7.9      | 12.4     |
|              | DirectMatchNet [10]    | 5.1      | 7.9      |
|              | Wormhole (Ours)        | **4.6**  | **6.7**  |

**Ablation study.** To evaluate which criterion, ours $\mathcal{C}_{\mathcal{W}}$ or the more conservative $\mathcal{C}_{\mathcal{T}}$ [12, 46], leads to better masks for partial shape correspondence, we compare masking strategies on the HOLES dataset in the following setting: Binary mask from $\mathcal{C}_{\mathcal{T}}$ [12, 46], non-binary mask from $\mathcal{C}_{\mathcal{T}}$ [12, 46], our binary mask, and our non-binary mask. As can be seen in Table 4, learning with masks derived from our wormhole criterion surpass the alternatives for partial shape correspondence, showing the importance of less restrictive criteria to find consistent distances between pairs of points.

## 5 Conclusion

In this paper, we analyzed invariant properties of geodesic distances between partial and full surfaces. Following it we developed an improved criterion for identifying *consistent* pairs – pairs where the geodesic distance remains consistent between the partial and full surfaces. Our improved criterion

Table 4: Ablation study on the criterion used to derive the mask in the proposed loss. We compare the criterion $\mathcal{C}_\mathcal{T}$ [12, 46] with the new wormhole $\mathcal{C}_\mathcal{W}$. Training and testing were conducted on the HOLES dataset. Numbers represent the average geodesic error ($\times 100$).

| Mask type | Binary | | Non-binary | |
|---|---|---|---|---|
| from | $\mathcal{C}_\mathcal{T}$ [12, 46] | $\mathcal{C}_\mathcal{W}$ (Ours) | $\mathcal{C}_\mathcal{T}$ [12, 46] | $\mathcal{C}_\mathcal{W}$ (Ours) |
| HOLES | 18.8 | **15.3** | 14.7 | **10.9** |

finds consistent pairs, or *guarantees* them, if the distance between two points on the partial surface is, for all couple of boundary points, smaller than the the sum of distances to the boundary points and of the distance in the embedding space between these boundary points, which is given by the straight line for Euclidean embeddings. We used this criterion to tackle the complex challenge of partial shape correspondence, by incorporating our guaranteed pairs in a novel unsupervised loss function specially designed to handle partial surfaces. With this loss function, our method demonstrated SOTA results on reference partial shape correspondence benchmarks, namely the SHREC'16 CUTS and HOLES and the recent PFAUST dataset.

**Limitations.** Our paper presents a novel criterion for finding consistent pairs in partial surfaces. However, it is still not able to recover all consistent pairs in general surfaces, meaning that some information that would be useful for partial shape matching is discarded. We suspect there could exist even less restrictive criteria for finding consistent geodesic distances in the presence of holes and cuts. The exploration of such criteria is left for future research. Additionally, finding boundaries in higher dimensions should be addressed with care, and operating in the embedding space in higher dimensions would not be as forgiving as what we have done for 2D surfaces in $\mathbb{R}^3$. In this work, we focused on partial-to-full shape matching. Although the wormhole criterion can be used to find consistent pairs on each shape independently, we would also need to find the shared consistent pairs for matching the shapes. Designing a full pipeline for partial-to-partial shape matching with our criterion is left for future work.

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

# A Further Theoretical Details and Proofs

## A.1 Proof of Theorem 1

*Proof.* The geodesic curve between the two points is a curve (in the embedding space) between the points restricted to the manifold $\mathcal{Y}$. The length of this curve is at least as long as that of the minimal geodesic (line in the Euclidean case) connecting the points in the embedding space. This trivial result holds since the metric on the manifold is the one induced by the metric of the (Euclidean) embedding space restricted to the embedded manifold. $\qquad\square$

## A.2 Proof of Theorem 2

*Proof.* Consider the first case where the shortest paths on the full surface $\mathcal{Y}$ from $P_{\mathcal{Y}}$ to $B_1$ and from $Q_{\mathcal{Y}}$ to $B_2$ do not intersect the boundary $\mathcal{B}$. In this case, we have that the computed geodesic distance on the partial surface to the boundary points is the same as that on the full surface. As such, $d_{\mathcal{Y}'}(P_{\mathcal{Y}}, B_1) = d_{\mathcal{Y}}(P_{\mathcal{Y}}, B_1)$ and $d_{\mathcal{Y}'}(Q_{\mathcal{Y}}, B_2) = d_{\mathcal{Y}}(Q_{\mathcal{Y}}, B_2)$. The shortest path on the full surface passing from $P_{\mathcal{Y}}$ to $Q_{\mathcal{Y}}$ passing through $B_1$ and $B_2$ will thus have length $d_{\mathcal{Y}}(P_{\mathcal{Y}}, B_1) + d_{\mathcal{Y}}(Q_{\mathcal{Y}}, B_2) + d_{\mathcal{Y}}(B_1, B_2) \geq d_{\mathcal{Y}}(P_{\mathcal{Y}}, B_1) + d_{\mathcal{Y}}(Q_{\mathcal{Y}}, B_2) + d_E(B_1, B_2)$ according to Theorem 1. Therefore if the pair satisfies the criterion, the geodesic distance computed on the partial surface is the geodesic distance on the full surface, meaning that the pair is consistent.

Consider now the second case where one or both of the shortest paths on the full surface $\mathcal{Y}$ from $P_{\mathcal{Y}}$ to $B_1$ and from $Q_{\mathcal{Y}}$ to $B_2$ intersect the boundary $\mathcal{B}$. Let $B_0$ and $B_3$ be the first intersection points on each of them. Any trajectory between them, such as the shortest path passing from $B_0$ to $B_1$ followed by the Euclidean straight line from $B_1$ to $B_2$ and then the shortest path from $B_2$ to $B_3$, are at least as long as the Euclidean straight line between $B_0$ and $B_3$: $d_E(B_0, B_3) \leq d_{\mathcal{Y}'}(B_0, B_1) + d_{\mathcal{Y}'}(B_2, B_3) + d_E(B_1, B_2)$. Since $B_0$ and $B_3$ are also closer to the original points, we then have $d_{\mathcal{Y}'}(P_{\mathcal{Y}}, B_0) + d_{\mathcal{Y}'}(Q_{\mathcal{Y}}, B_3) + d_E(B_0, B_3) \leq d_{\mathcal{Y}'}(P_{\mathcal{Y}}, B_1) + d_{\mathcal{Y}'}(Q_{\mathcal{Y}}, B_2) + d_E(B_1, B_2)$. Since the pair $B_0$ and $B_3$ satisfy the first case, the criterion will validate the pair $P_{\mathcal{Y}}$ and $Q_{\mathcal{Y}}$ only if it is consistent.

$\qquad\square$

## A.3 Generalisation to Non-Standard Metrics

So far, we implicitly used the fact that we were working with the standard way of computing geodesic distances on surfaces. Formally, this means that we only considered manifolds equipped with the standard uniform isotropic Riemannian metric, where unit steps in the tangent plane have the same length as unit steps in the embedding space. Schematically, the manifold and the embedding space shared the same ruler. This allowed us to bound the Riemannian length of the geodesic curve with the Euclidean length of the straight line in the embedding space between points.

However, in some cases, it is natural or beneficial to consider other metrics on the manifold. As new metrics change the length concept, this means that a same curve on the manifold will have a different length. With other metrics, the manifold and the embedding space no longer share the same ruler, and unit steps on the manifold tangent space are no longer of unit Euclidean length. We must thus adapt our criterion in consequence.

A Riemannian metric on the manifold is defined by a symmetric definite positive matrix $M$, such that at any point $x \in \mathcal{X}$, the length of the tangent vector $u$ at point $x$ is given by a quadratic form $\mathcal{R}_x(u) = \sqrt{u^\top M(x)u}$. The geodesic length of a curve on the surface is given by integrating the Riemannian length of all the infinitesimal tangent steps: $\int_0^1 \mathcal{R}_{\gamma(t)}(\gamma'(t))dt$ where $\gamma(t)$ parametrises the curve monotonically. The link between the manifold Riemannian length and the embedding Euclidean length of steps is given by the eigenvalues of the metric $M$. Let $\mu_1(x) \leq \mu_2(x)$ be its smallest and largest eigenvalues, with eigenvectors $v_1(x)$ and $v_2(x)$ of unit Euclidean norm $\|v_1(x)\|_2 = \|v_2(x)\|_2 = 1$. Then Euclidean unit steps have length bounded in $[\sqrt{\mu_1(x)}, \sqrt{\mu_2(x)}]$. Under the assumption that $\min_x \mu_1(x) = C_M > 0$, which systematically occurs for common metrics on natural surfaces, Euclidean unit steps have at least $\sqrt{C_M}$ Riemannian length. As such, reparametrising the curve with the standard Euclidean arclength, i.e. the arclength for the Riemannian metric with matrix $I$, we can bound each infinitesimal step along the curve, which has a fixed

Euclidean length, to get a bound on the geodesic length: $d_M(x, y) \geq \sqrt{C_M} d_I(x, y)$, where $d_M$ is the geodesic distance on the Riemannian manifold associated to the matrix $M$.

We return to partial surfaces, with $\mathcal{Y}$ and $\mathcal{Y}'$ full and partial versions of the same surface. The surfaces are associated with the intrinsic Riemannian metric $M$. As such, geodesic distances are changed to $d_{\mathcal{Y}} = d_M$ and are no longer necessarily equal to $d_I$. We assume that $C_M$ is known[2]. We can now adapt our wormhole criterion to the Riemannian metric $M$ to define the new wormhole metric-sensitive criterion

$$\mathcal{C}_\mathcal{W}(P_\mathcal{Y}, Q_\mathcal{Y}) = \begin{cases} 1 & \text{if } d_{\mathcal{Y}'}(P_\mathcal{Y}, Q_\mathcal{Y}) \leq \min_{B_1, B_2 \in \mathcal{B}} d_{\mathcal{Y}'}(P_\mathcal{Y}, B_1) + d_{\mathcal{Y}'}(Q_\mathcal{Y}, B_2) + \sqrt{C_M} d_E(B_1, B_2), \\ 0 & \text{otherwise.} \end{cases}$$
(9)

Note, that in the degenerate case $C_M = 0$, our criterion $\mathcal{C}_\mathcal{W}$ degenerates to $\mathcal{C}_\mathcal{T}$ as we are unable to bound path lengths inside missing parts. By design, this generalised criterion to Riemannian metrics can be used to find consistent pairs.

**Theorem 3** ($\mathcal{C}_\mathcal{W}$ guarantees). *The metric-sensitive wormhole criterion $\mathcal{C}_\mathcal{W}$ yields guaranteed pairs.*

*Proof.* The proof generalises the one of Theorem 2. The difference comes when comparing the length of geodesic paths on the full surface $\mathcal{Y}$ to length of curves in the embedding space. Name $d_I$ the *curved $M$-based Euclidean distance* of geodesic curves on the manifold for the metric $M$. The curved $M$-based Euclidean distance is greater than the Euclidean distance of straight lines between the points $d_I \geq d_E$ (Theorem 1). Since $d_M \geq \sqrt{C_M} d_I$, i.e. the geodesic distance is at least $\sqrt{C_M}$ times larger than the curved $M$-based Euclidean distance, we have the relationship between the geodesic and Euclidean distances $d_\mathcal{Y} \geq \sqrt{C_M} d_E$. We can then conclude by applying the same reasoning as in the proof of Theorem 2. $\square$

# B  Further Experimental Details

## B.1  Multi-Dimensional Scaling

Formally, the original MDS problem aims to minimise the quadratic stress function

$$X^* = \underset{X \in \mathbb{R}^{n \times |V|}}{\arg\min} \sum_{i,j} w_{ij} (d_{\mathbb{R}^n}(X_i, X_j) - d_\mathcal{Y}(P_i, P_j))^2,$$
(10)

where $w_{ij} \in [0, 1]$ is a given optional weighting scheme. A common assumption in the theory of MDS is that the given manifold can indeed be isometrically mapped to a low dimensional Euclidean space[3]. If this assumption holds we could consider all pairs equally $w_{ij} = 1$, which yields the classical scaling approach [52, 57].

However, we would like embeddings of partial surfaces $\mathcal{Y}'$ to be robust to missing parts, which implies that we need to handle the boundary conditions induced from partial surfaces. Enforcing the preservation of distances for inconsistent pairs could distort the embedding. To overcome the influence of holes, cuts, and boundaries in general, we resort to our criterion to filter out inconsistent pairs.

In [12, 14, 46, 47], the weights are set to $w_{ij} = \mathcal{C}_\mathcal{T}(P_i, P_j)$, leading to the TCIE method [46, 47]. To the best of our knowledge, [46, 47] provides the best weighting scheme based on consistent pairs of points without involving heuristics. Other approaches exist relying on heuristics, such as focusing on local pairs of points due to local approximations of geodesic distances [51]. We propose to refine the TCIE idea by taking a more inclusive criterion instead $w_{ij} = \mathcal{C}_\mathcal{W}(P_i, P_j) = \boldsymbol{M}_{ij}$. Minimising Equation (10) can be efficiently done by any optimization method, for example, the SMACOF algorithm [9] was shown to be equivalent to a constant step size weighted gradient descent [17]. Classical scaling [52, 57] is a method of choice for initialization to avoid local minima.

Our toy surfaces are pointclouds of swiss rolls with 2000 randomly sampled vertices, stretched in width by a factor of 1.5, to which either no noise (Figure 7) or very small Gaussian noise of

---

[2]For full-to-partial shape matching, then $C_M$ can be computed on the full surface. For partial-to-partial shape matching, we assume that $C_M$ is given by an oracle.

[3]This assumption is obviously violated for surfaces with effective Gaussian curvature, as can be easily verified by the Gauss-Bonnet theorem [8].

standard deviation $0.2$ is added[4] (Figure 8). For each surface, we removed either a hole or a missing part connected to the boundary. We use Dijkstra's algorithm to compute geodesic distances on the pointclouds. Following [46, 47], we also set $w_{ij} = 1$ for small geodesic distances (smaller than 3) to handle the boundaries. We here compare with more reference works in manifold learning than in the main manuscript, namely Isomap [57], LLE [49], Hessian LLE [23], MLLE [62], LTSA LLE [63], Laplacian eigenmaps [5], Diffusion maps [19], t-SNE [58], UMAP [37] PCA [42, 26], Kernel PCA [50], and TCIE [46, 47]. We use either the Euclidean or geodesic distance matrix for Diffusion Maps, t-SNE, and UMAP. We run t-SNE with perplexity parameter of $50$ and UMAP with minimum distance parameter of $0.8$. For both methods, these high parameter values are supposed to respect data topology rather than encourage clustering. Methods requiring nearest neighbor computations use $k = 15$ neighbors. For reference, we also show the computed embeddings on the full surfaces.

In the ideal noiseless case, the non-vanilla LLE methods are able to handle the partial surface with a hole quite well. However, only a minor amount of noise breaks these methods completely. The t-SNE and UMAP methods perform poorly, even on the full surfaces. These popular methods were indeed primarily designed to show nice clusters, but are not intended and should not be used for viewing non-clustered data manifolds in low dimensions as is too often the case. Both TCIE and our WHCIE are robust to small amounts of noise, yet our embeddings are superior as they are less distorted by the missing parts as we find more consistent pairs.

## B.2 Discretization

The definition of consistent pairs is so far in a continuous setting. We need to be adapt it to the discrete world, as computational surfaces are themselves discrete. Let $V$ be a finite set of vertices sampled from the continuous surface. We only compute guaranteed pairs between the vertices of $V$. To that end, we consider only the discrete set of boundary vertices $B = V \cap \mathcal{B}$. Given this discretization, we calculated $\mathcal{C}_{\mathcal{W}}$-guaranteed pairs in a naive manner. We first compute the geodesic distance matrix $\boldsymbol{D}_{\mathcal{Y}'}$, having $(i, j)$ entry $(\boldsymbol{D}_{\mathcal{Y}'})_{ij}$ as the geodesic distance on the partial surface $\mathcal{Y}'$ between the $i$-th and $j$-th vertices of $V$, using a relevant method for computing geodesic distances on the discrete surface, e.g. Dijkstra's algorithm [21], Fast Marching Method [28], MMP [56, 39], or deep learning methods for geodesic distance calculation [32, 27]. We then calculate the Euclidean distance between each pair of boundary vertices in $B$. We have thus computed all elements of the threshold in Equation (3), and all that is left is to search for the two boundary vertices that give the minimal bound. Optionally, computations can be sped up using a GPU, we used a V100 and it took a few minutes to compute the masks for surfaces on the PFAUST datasets. We found that it is best to compute the $\mathcal{O}(|V|^2|B|^2)$ distances in batches due to the high spatial complexity, which for common surfaces becomes prohibitively large. This process computes a matrix $\boldsymbol{K}$ of size $|V| \times |V|$, containing the threshold of the criterion $\mathcal{C}_{\mathcal{W}}$ calculated for each pair of vertices. The guaranteed pairs are the true elements of the binary mask $\boldsymbol{M}_{ij} = \mathbb{1}_{(\boldsymbol{D}_{\mathcal{Y}'})_{ij} \leq \boldsymbol{K}_{ij}}$.

An alternative faster approach to the naive method above for computing the mask matrix $\boldsymbol{M}$ would be to recompute geodesic distances on a tweaked distance graph. The adjustment would be to create an edge between all boundary points with distance equal to the Euclidean distance between them. Shortest paths on this tweaked graph belong to one of two types. The first type of new shortest paths between any two points corresponds to paths that do not pass through boundary points, which means that they are the same as the ones on the partial surface $\mathcal{Y}'$: the pair is consistent and guaranteed by $\mathcal{C}_{\mathcal{W}}$. The other type of new shortest paths between any two points corresponds to paths using a single new edge between boundary vertices, which means that their length is equal to the wormhole worst case distance bound between these two points: the pair is not guaranteed by $\mathcal{C}_{\mathcal{W}}$. This procedure thus computes the criterion matrix $\tilde{\boldsymbol{K}}$ defined as

$$\tilde{\boldsymbol{K}}_{ij} = \min\{(\boldsymbol{D}_{\mathcal{Y}'})_{ij}, \boldsymbol{K}_{ij}\}. \tag{11}$$

We can then compute the binary mask matrix by comparing the distances computed on the original and the tweaked graphs $\boldsymbol{M}_{ij} = \mathbb{1}_{(\boldsymbol{D}_{\mathcal{Y}'})_{ij} \leq \tilde{\boldsymbol{K}}_{ij}}$. The total cost of this approach comes from computing twice the distance algorithm, once on a graph and once on its modification with $\Theta(|B|^2)$ extra edges. As such, the complexity of this method is $\Theta\big(|V|\big(|E| + |B|^2 + |V|\log(|V|)\big)\big)$. However, in practice, we usually have $|B| = O(\sqrt{|V|})$. As such, the complexity is usually $O\big(|V|\big(|E| + |V|\log(|V|)\big)\big)$,

---

[4]We provide the code to generate the pointclouds in our code repository.

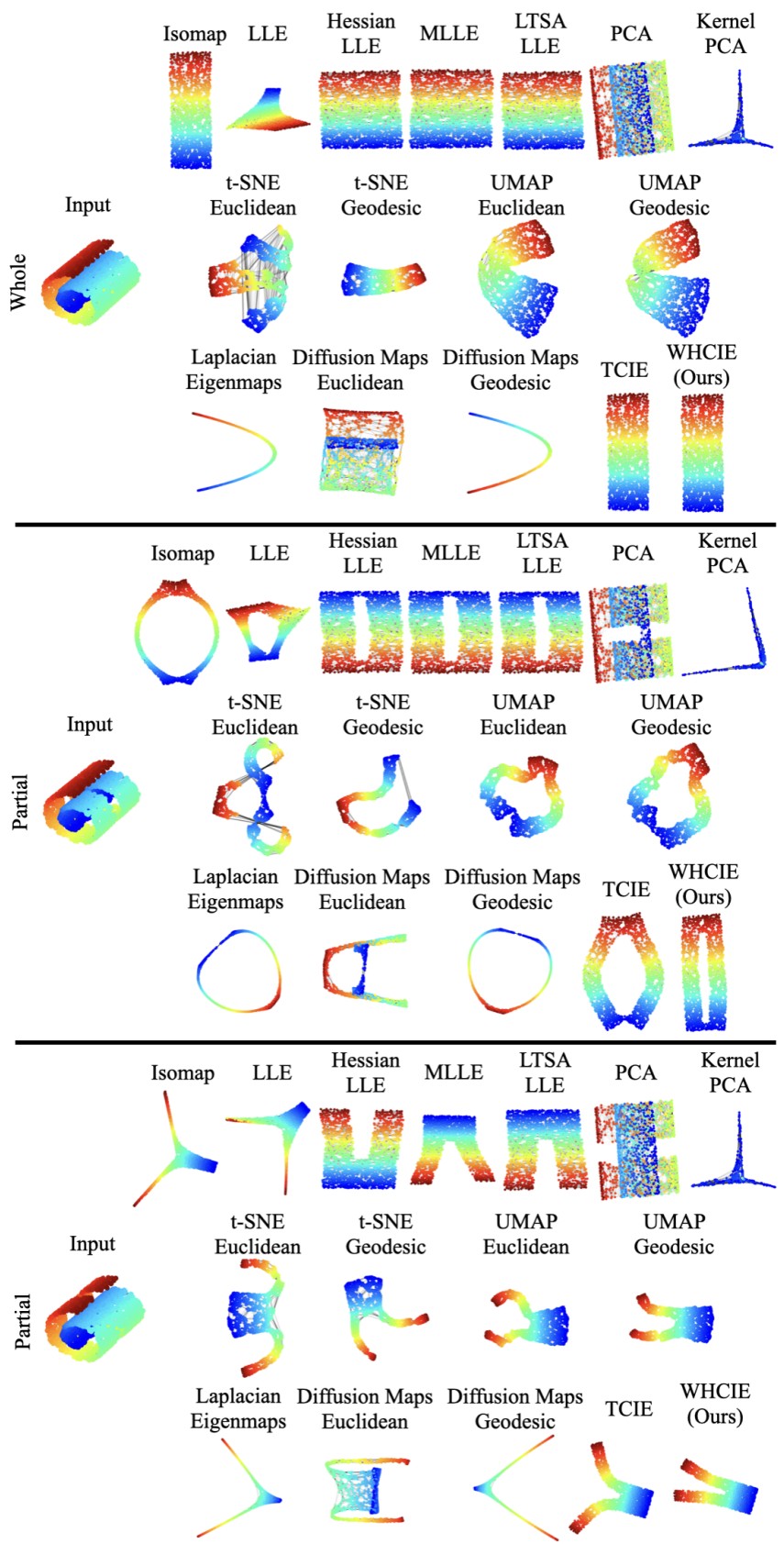

Figure 7: MDS embeddings of reference manifold learning methods on a whole swiss roll without noise and its variants with a hole or a cut.

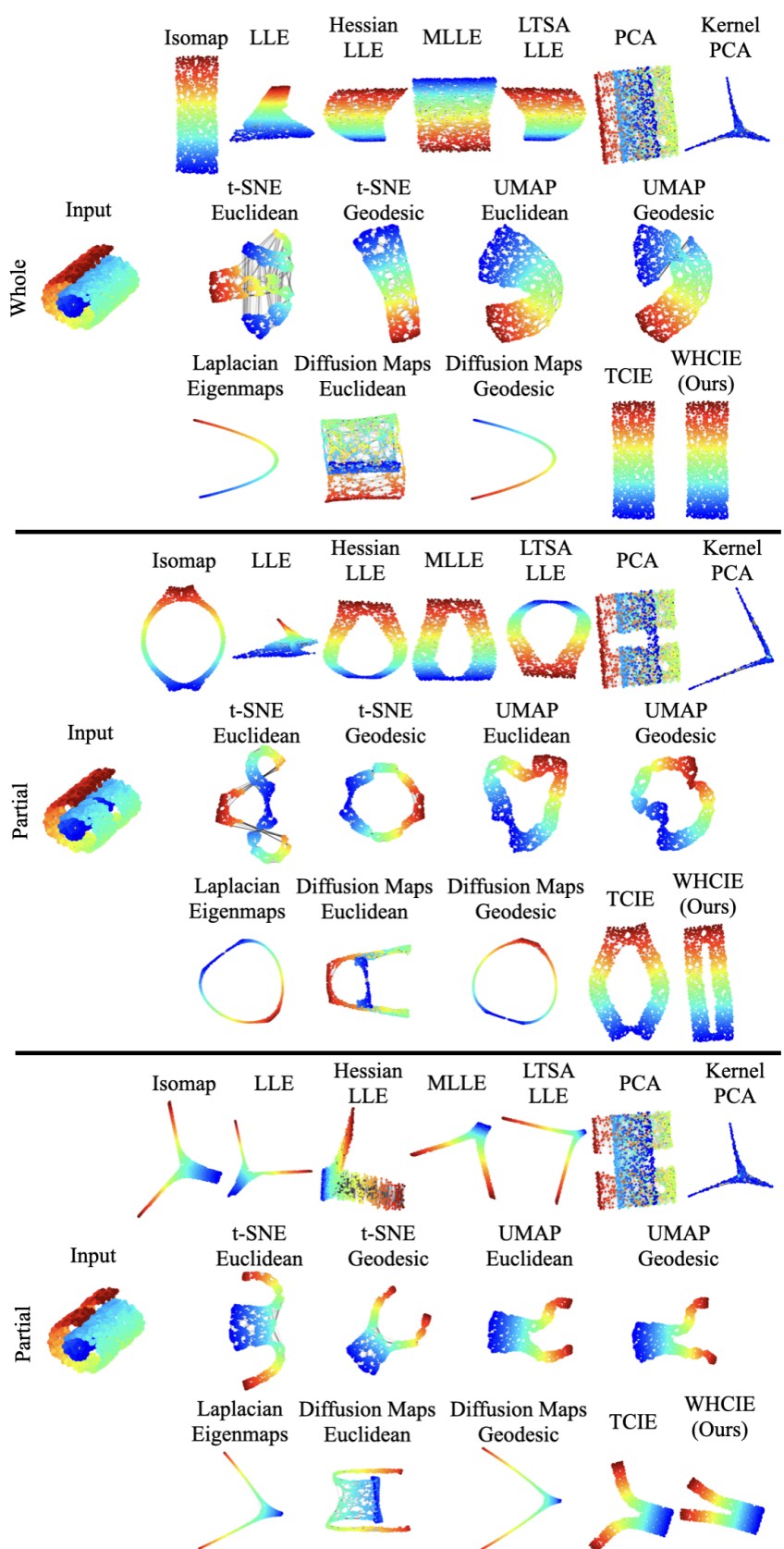

Figure 8: MDS embeddings of reference manifold learning methods on a whole swiss roll with small noise and its variants with a hole or a cut.

meaning that computing the mask criterion $\boldsymbol{K}$ has the same complexity as computing the shortest distances on the partial shape $\boldsymbol{D}_{\mathcal{Y}'}$. Note that for triangulations or kNN graphs, $|E| = \Theta(|V|)$ and then the complexity for computing the wormhole mask is simply $O\big(|V|^2 \log(|V|)\big)$ in practice.

## B.3 Derivation of the Masked Geodesic Distance Preservation Loss

Our discrete loss function $\mathcal{L}_{\text{geo}}$ in Equation (7) is based on the continuous loss presented in [1], see also [13]. Following their formulations we introduce a loss for the correspondence function between $\mathcal{X}$ and $\mathcal{Y}'$, defined by $\tilde{p} : \mathcal{Y}' \times \mathcal{X} \to \mathbb{R}^+$ as,

$$\int_{\mathcal{Y}' \times \mathcal{Y}'} \left( \int_{\mathcal{X} \times \mathcal{X}} d_{\mathcal{X}}(x_1, x_2) \tilde{p}(x_1, y_1') \tilde{p}(x_2, y_2') da_{x_1} da_{x_2} - d_{\mathcal{Y}'}(y_1', y_2') \right)^2 m(y_1', y_2') da_{y_1'} da_{y_2'} \quad (12)$$

where $d_{\mathcal{X}}$ and $d_{\mathcal{Y}'}$ measure the distances between surface points, and $m : \mathcal{Y}' \times \mathcal{Y}' \to \{0, 1\}$ is our binary masking function. We readily have the discrete version,

$$\mathcal{L}(\tilde{\boldsymbol{P}}, \boldsymbol{D}_{\mathcal{X}}, \boldsymbol{D}_{\mathcal{Y}'}, \boldsymbol{M}) \quad = \quad \|\sqrt{\boldsymbol{M}} \odot (\tilde{\boldsymbol{P}} \boldsymbol{A}_{\mathcal{X}} \boldsymbol{D}_{\mathcal{X}} \boldsymbol{A}_{\mathcal{X}} \tilde{\boldsymbol{P}}^{\top} - \boldsymbol{D}_{\mathcal{Y}})\|_{\mathcal{Y}' \mathcal{Y}'}, \quad (13)$$

where $\|\boldsymbol{B}\|_{\mathcal{Y}' \mathcal{Y}'} = \text{trace}(\boldsymbol{B}^{\top} \boldsymbol{A}_{\mathcal{Y}} \boldsymbol{B} \boldsymbol{A}_{\mathcal{Y}})$ and $\sqrt{\boldsymbol{M}}$ is the entry-wise square-root of $\boldsymbol{M}$. Similar to previous papers [25] we assume that our learned correspondence matrix implicitly contains the area matrix $\boldsymbol{A}_{\mathcal{X}}$, thus, $\boldsymbol{P} = \tilde{\boldsymbol{P}} \boldsymbol{A}_{\mathcal{X}}$. Consequently, our loss function is defined as follows,

$$\mathcal{L}_{\text{geo}}(\boldsymbol{P}, \boldsymbol{D}_{\mathcal{X}}, \boldsymbol{D}_{\mathcal{Y}'}, \boldsymbol{M}) \quad = \quad \|\sqrt{\boldsymbol{M}} \odot (\boldsymbol{P} \boldsymbol{D}_{\mathcal{X}} \boldsymbol{P}^{\top} - \boldsymbol{D}_{\mathcal{Y}'})\|_{\mathcal{Y}' \mathcal{Y}'}. \quad (14)$$

For simplicity, we expressed it element-wise,

$$\mathcal{L}_{\text{geo}}(\boldsymbol{P}, \boldsymbol{D}_{\mathcal{X}}, \boldsymbol{D}_{\mathcal{Y}'}, \boldsymbol{M}) \quad = \quad \sum_{ij} \boldsymbol{M}_{ij} \boldsymbol{A}_{\mathcal{Y}' ii} \boldsymbol{A}_{\mathcal{Y}' jj} \left( (\boldsymbol{P} \boldsymbol{D}_{\mathcal{X}} \boldsymbol{P}^{\top} - \boldsymbol{D}_{\mathcal{Y}'})_{ij} \right)^2. \quad (15)$$

The same derivation led to the non-masked loss in [10].

## B.4 Detailed Implementation Considerations

Most partial shape matching pipelines [3, 18, 10] can be divided into four parts; feature refinement network, correspondence matrix creation, loss functions, and post processing. The feature refinement network gets raw input features of the surface, such as $xyz$ coordinates, and additional geometric properties of the surface, e.g. the LBO eigendecomposition, and outputs a refined feature for each vertex. Given the refined features from each surface, a correspondence matrix is built either from FM [48, 25], or, as in our pipeline, directly from similarities between features on the different surfaces [2, 18, 10]. The loss functions operate on the FM matrix that is calculated directly from the feature matrix [48], or on the FM which is calculated from the correspondence matrix [10], or on the correspondence matrix directly [25]. Post processing allows to refine test time results and usually consists in applying the methods Zoom-out[38], PMF [60], or refinement of the network weights via test time adaptation [18].

We use as input features the $xyz$ coordinates of each vertex along with its estimated normal. We took DiffusionNet [54] to be the feature extraction network, with shared weights between the full and the partial surfaces. It outputs the refined features $\boldsymbol{F}_{\mathcal{S}} \in \mathbb{R}^{|V_S| \times d}$ where $d = 256$ is the feature dimensionality, and $\mathcal{S} \in \{\mathcal{X}, \mathcal{Y}'\}$. We follow [11], by replacing the FM layer with a direct computation of the correspondence matrix via Softmax similarity, similar to [7, 18],

$$\boldsymbol{P}_{ij} = \frac{\exp(\boldsymbol{G}_{ij}/\tau)}{\sum_i \exp(\boldsymbol{G}_{ij}/\tau)}, \quad (16)$$

where $\boldsymbol{G} = \boldsymbol{F}_{\mathcal{Y}'}^{\top} \boldsymbol{F}_{\mathcal{X}}$, and $\tau$ is a temperature hyper-parameter, where $\tau = 0.07$ for the HOLES dataset and $\tau = 0.01$ for the CUTS and PFAUST datasets. We point out that the correspondence matrix is row normalised $\sum_i \boldsymbol{P}_{ij} = 1$ but not column normalised, as some vertices of $\mathcal{X}$ in missing parts of $\mathcal{Y}$ should not have corresponding vertices. The correspondence matrix is then fed into our loss function $\mathcal{L}_{\text{geo}}$ using the soft mask $\boldsymbol{M}^{\tilde{s}}$ unless stated otherwise (Equation (7)), with Lagrangian coefficient $\lambda_{\text{geo}} = 10^3$. Following [48], we additionally use regularization based on the FM matrix $\boldsymbol{C}$ calculated directly from the correspondence matrix $\boldsymbol{P}$,

$$\boldsymbol{C} = \boldsymbol{\Phi}_{\mathcal{Y}'}^{\top} \boldsymbol{A}_{\mathcal{Y}'} \boldsymbol{P} \boldsymbol{\Phi}_{\mathcal{X}}, \quad (17)$$

where $\mathbf{\Phi}_{\mathcal{X}}$ and $\mathbf{\Phi}_{\mathcal{Y}'}$ are the LBO eigenfunctions of the surfaces $\mathcal{X}$ and $\mathcal{Y}'$, and $\mathbf{A}_{\mathcal{Y}'}$ is diagonal matrix containing the area associated to each vertex in the discrete surface $\mathcal{Y}'$. We plug the FM matrix into the regularisation loss $\mathcal{L}_{\text{ortho}}$ with Lagrangian coefficient $\lambda_{\text{ortho}} = 1$, which promotes the preservation of vertex area under the estimated correspondence [48, 3],

$$\mathcal{L}_{\text{ortho}}(\mathbf{C}) = \left\| \mathbf{C}\mathbf{C}^{\top} - \mathbf{J}_r \right\|_F, \tag{18}$$

where $\mathbf{J}_r = \left[ \begin{smallmatrix} \mathbf{I}_r & \mathbf{0} \\ \mathbf{0} & \mathbf{0} \end{smallmatrix} \right]$ is the identity matrix until column $r$ and only zeros columns afterwards, and $r$ is number of eigenvalues of the LBO on $\mathcal{Y}'$ that are smaller than the largest eigenvalue of the truncated LBO on the discrete surface $\mathcal{X}$. We trained our network for 20000 iterations, with Adam optimizer [29], with a learning rate of $10^{-3}$ and a cosine annealing scheduler [35] with minimum learning rate parameter $\eta_{min} = 10^{-4}$ and maximum temperature of $T_{max} = 300$ steps. Lastly, for post-processing, we use the test time adaptation refinement method [18], that refines the network weights separately for each pair of surfaces with 15 iterations of gradient descent.

Compared to DirectMatchNet [10], our method only involves an additional preprocessing of the shape to compute the mask matrices for the loss function. As such, from a time perspective, both methods are comparable and even identical at inference time without test time adaptation. For reference, inference time is 0.12s on average on our hardware for both DirectMatchNet and our method on the HOLES dataset.

## B.5 Further Shape Correspondence Result

We here display further qualitative results on the PFAUST M (Figure 9), H (Figure 10), SHREC'16 CUTS (Figure 11), and HOLES (Figure 12). The results support those presented in Figure 6. Methods not based on the preservation of distances produce significant distortions compared to those that do. Furthermore, by focusing on consistent pairs, our method yields less distortion near holes than a similar approach [10] not taking them into account.

We also provide in Figure 13 Percentage of Correct Keypoint (PCK) curves describing the percentage of correct predictions with respect to the geodesic error for the baseline methods UnsupDPFM [3], RobustFMnet [18], DirectMatchNet [10], and our method, without refinement for all methods. In particular, our method yields clear improvement on the challenging PFAUST-H benchmark, and moderate improvements on the other datasets, which corroborates the results of Tables 2 and 3.

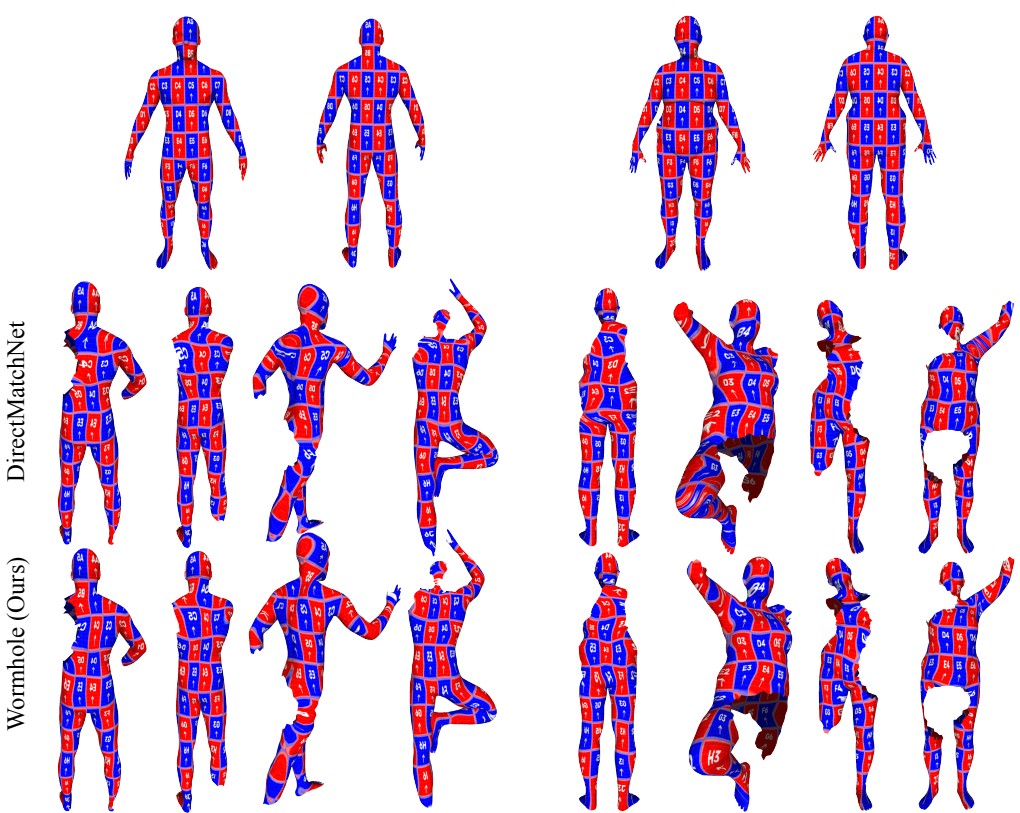

Figure 9: Additional qualitative results on the PFAUST-M dataset.

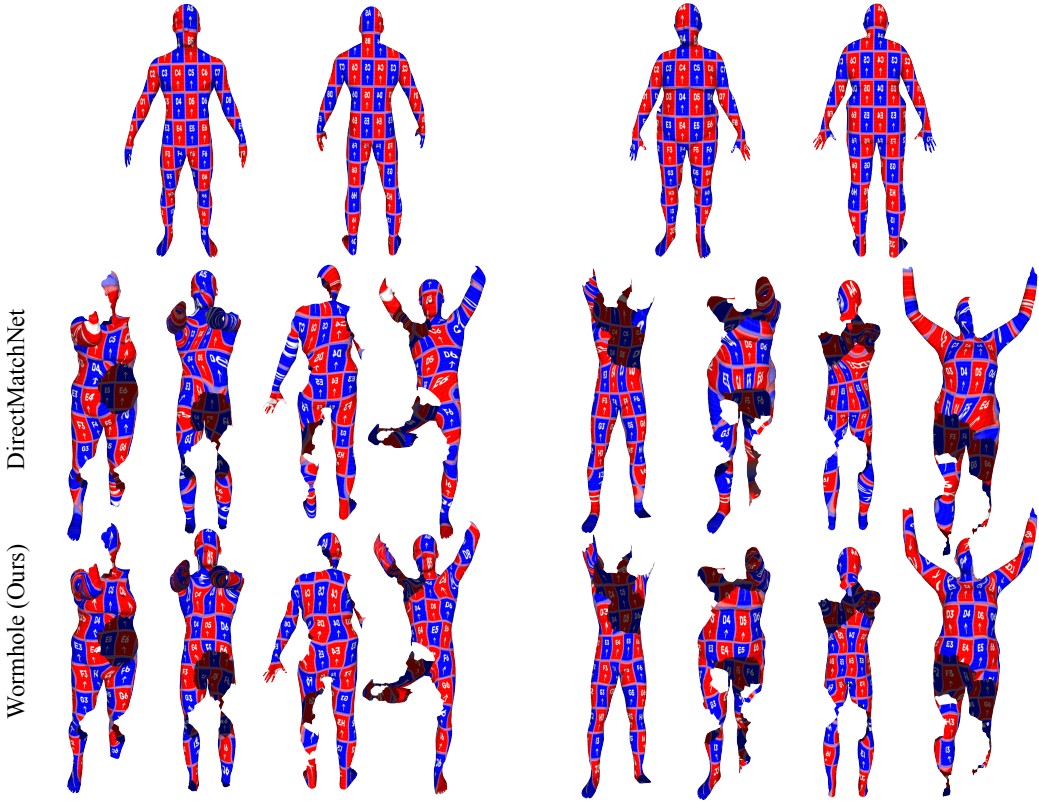

Figure 10: Additional qualitative results on the PFAUST-H dataset.

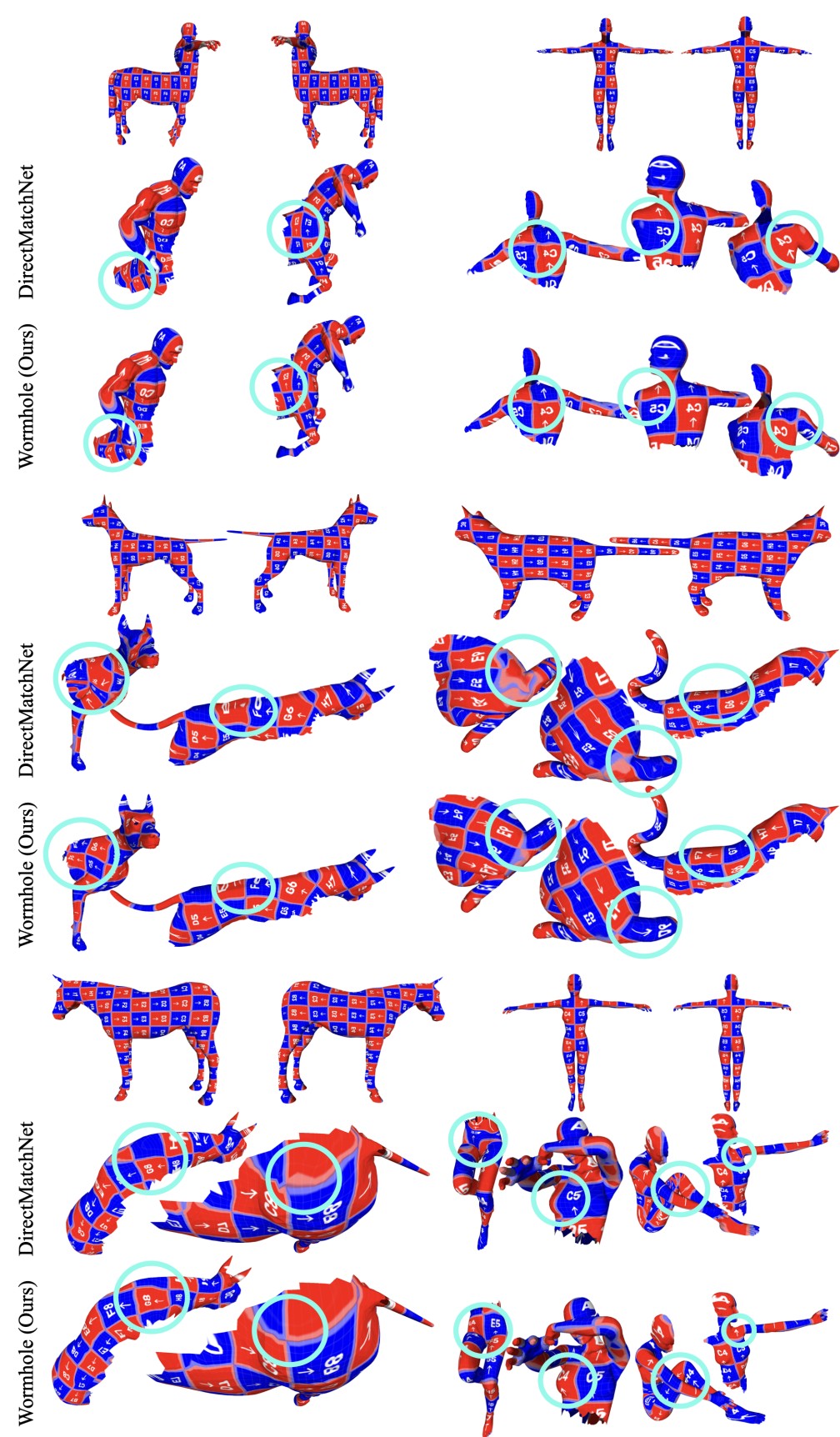

Figure 11: Additional qualitative results on the SHREC'16 CUTS dataset.

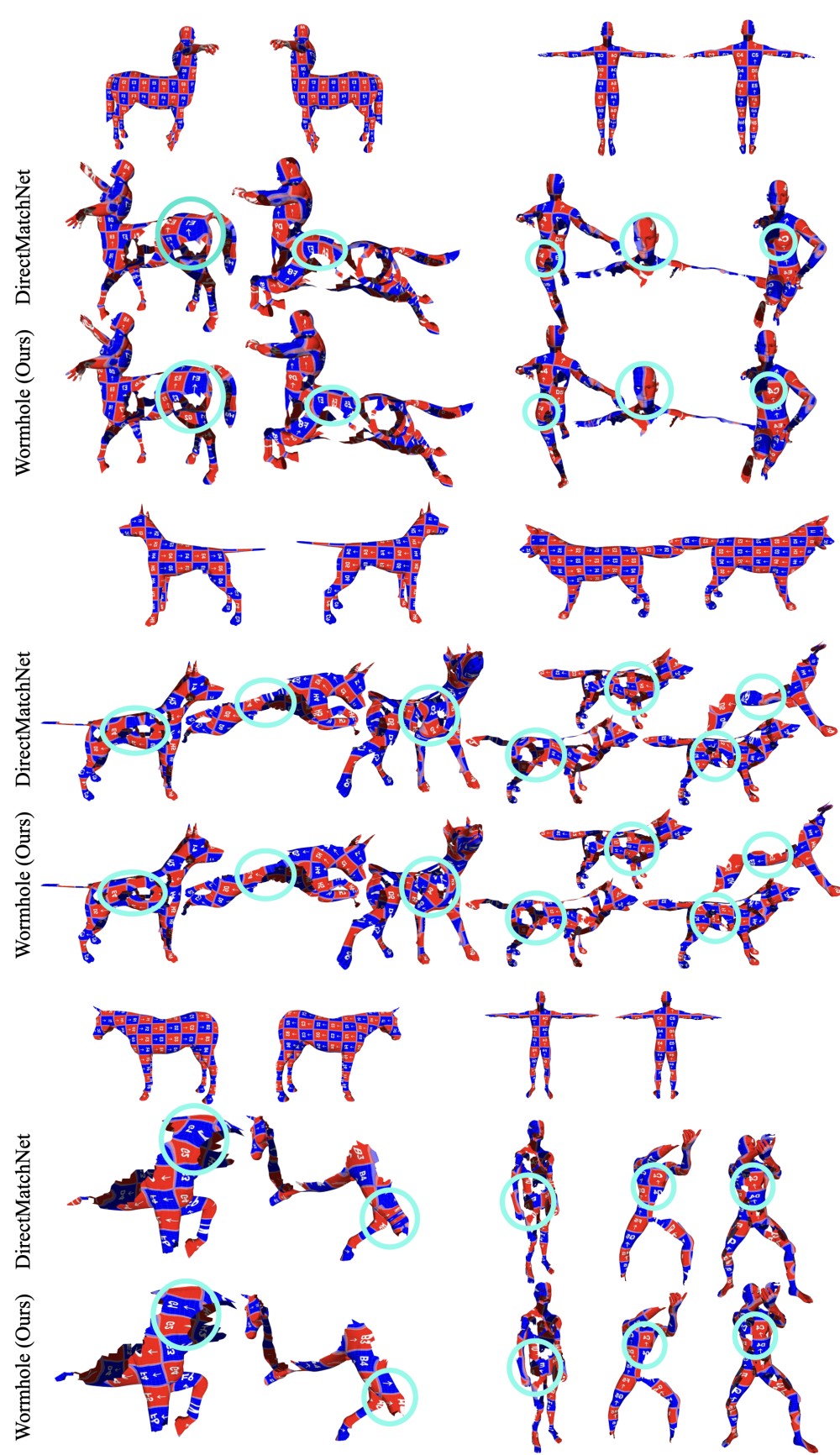

Figure 12: Additional qualitative results on the SHREC'16 HOLES dataset.

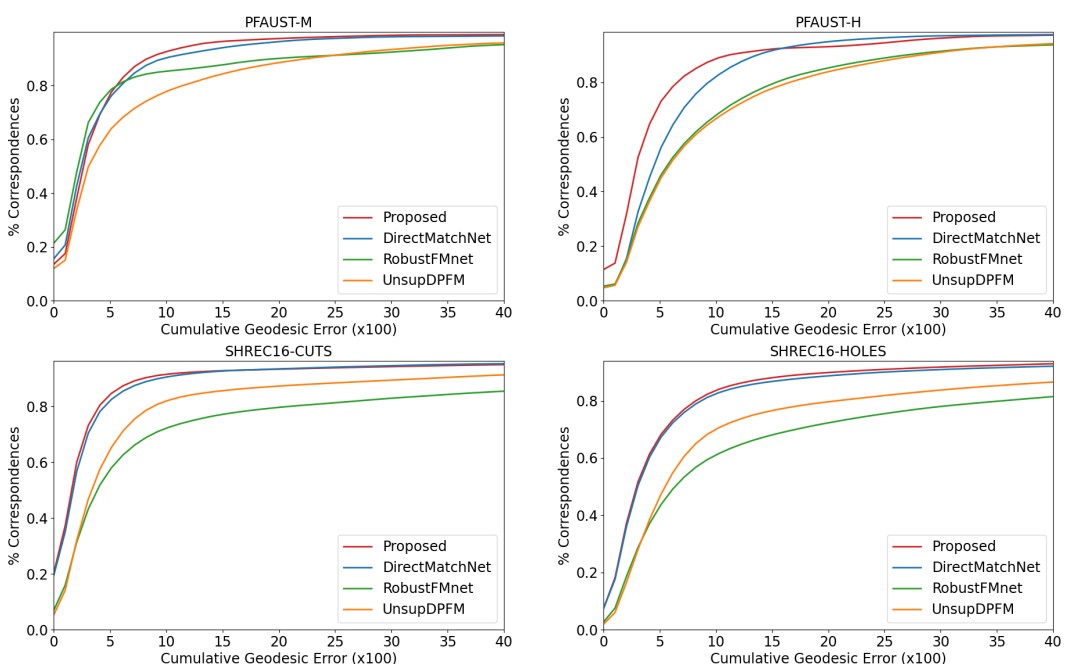

Figure 13: PCK curves of the geodesic error of the baselines and our method on the SHREC'16 CUTS and HOLES and the PFAUST M and H datasets.

