# OpenReview forum: "Wormhole Loss for Partial Shape Matching"
_NeurIPS.cc/2024/Conference — NeurIPS 2024 poster_

### Official Review · Reviewer_a2Wg · 2024-07-08

**Soundness:** 3
**Presentation:** 2
**Contribution:** 2
**Rating:** 5
**Confidence:** 4

**Summary:**

The paper proposes an unsupervised optimization for partial shape matching between (near-)isometric shapes, relying on distance similarity. Defined as the "Consistent Pairs" a pair of points that present a similar geodesic distance between the full and the partial shapes, this work relaxes this definition by assuming that the path on the partial shapes can traverse holes with straight lines. The paper shows results on CUTS and HOLES on par with the state-of-the-art and a slight improvement on PFAUST-M and PFAUST-H.

**Strengths:**

- The paper addresses a relevant and challenging problem: solving for partial shape correspondence in an unsupervised setting is a difficult task, and shape matching community is active on this
- The method shows interesting applicability in MDS embeddings, although the paper does not evaluate this aspect numerically.

**Weaknesses:**

- The proposed principle seems sound but is also restricted to simple cases. The method shows better results on HOLES, where the assumption holds, while for CUTS, it seems to suffer. My intuition is that CUTS contains a large missing part, and this cannot be well approximated by straight lines on the boundary.
- Some aspects of the presentation are not satisfactory. I appreciate the effort in grounding the formulation in a more systematic methodological explanation, but at the moment, it seemed cluttered. For example, I would say that Theorem 1 and its demonstration are unnecessary. Visualizations are not good for black-and-white printing. Figure 6 misses a Ground Truth to assess the quality of the final result.
- The paper and its limitations do not offer a discussion nor analysis on the limit of the proposed strategy in terms of failure cases. A more in-depth analysis of the impact of partiality magnitude and its nature on the capability of finding guaranteed pairs would be beneficial and insightful for future work.

**Questions:**

1) The paper analyzes the cases in which the partial shape is still a single component. How does the method perform in the presence of disconnected components?
2) How computationally expensive is the method?

**Limitations:**

The authors describe some limitations regarding future works but do not discuss failure cases of the proposed strategy.

---

> ### Author Rebuttal · Authors · 2024-08-06
>
> We  thank the reviewer for the detailed comments and for acknowledging the relevance and challenge of addressing partial shape correspondence in an unsupervised setting, as well as the applicability of  our method  to MDS embeddings. We will add to Figure 6 the ground-truths. We hereby respond to the reviewer's remarks and questions.
>
> **Clarification:** Regarding the reviewer's summary, it seems that there might have been some confusion and mix of concepts that we would like to address. As defined in the paper, consistent pairs are pairs of points for which the shortest path on the partial shape have rigorously the same length as the shortest path on the full shape. The lengths can be different only if the shortest path on the full shape passes through parts which are missing in the partial shape. We design a criterion that finds as many consistent pairs as possible. All pairs guaranteed by this criterion are definitely consistent. Our criterion is based on a lower bound estimation of the unknown geodesic distance on the full shape. It is realized by considering Euclidean distances between boundary points of the partial shape as potential shortcuts.
> ### CUTS
> Regarding performance on CUTS, the reviewer suggests that the proposed criterion is restricted to simple cases, and would not apply to missing parts like in CUTS due to their large size. We respectfully disagree with these two claims as explained below.
>
> - Regarding simplicity, "The missing parts in the CUTS dataset were created by slicing figures with 3D planes. In practice, this procedure does not modify much the topology of the surface and it does not introduce many inconsistent pairs. As such, cuts mostly preserve all geodesic distances, thus, our novel loss function was almost identical to the one in Equation (4)" (lines 256-258). The implication is that cuts provide too simple, and not too complex, partial shapes where almost are pairs are consistent and found by our criterion, explaining why there our method does not lead to significant improvement.
>
> - Regarding size, it is the shape of the missing parts, and not their size, that dictates the complexity of the partial shape. For example, cutting off just a finger or both legs on a human shape leads to simple partial shapes with only consistent pairs but with very different size of missing parts.
>
> In contrast, holes of various sizes present significantly harder topologies with many pairs that are inconsistent. In fact, our method leads to significant improvements when handling missing parts with holes by filtering out inconsistent pairs, as shown on the HOLES, PFAUST-M and PFAUST-H datasets.
> ### Failure cases
> Regarding failure cases of our criterion. Our binary criterion cannot guarantee inconsistent pairs as proven in Theorem 2. Regarding failure cases of our methods, in all our experiments except on CUTS, our criterion managed to find a significant amount of consistent pairs, leading to improved performance in both shape matching and manifold flattening. In CUTS, our criterion does not fail, it is just not so relevant as almost all pairs are consistent, and we discuss this issue in the results section. At the other end, we do not claim that our regularized approach is guaranteed to always succeed.
> ### Partiality magnitude and amount of missed consistent pairs
> The reviewer suggests to analyse the impact of partiality magnitude on the ability to find consistent pairs. This question is interesting, however, we would like to emphasize that the amount of missing parts is not the main factor impacting consistent pairs, but more the topology of the missing parts. See our earlier discussion on CUTS for an example of a simple partial shape with large missing parts. Nevertheless, in the rebuttal period, we evaluated the gap between the total amount of consistent pairs and that found by different criteria on both the PFAUST-M and PFAUST-H datasets, where PFAUST-H is harder since it has many more holes (of smaller size). See the table below, that we will include in the paper, presenting the percentage of consistent pairs, and among those the percentage of guaranteed pairs by the different criteria, along with the standard deviations.
> ||%Consistent|%Guaranteed ($\mathcal{C}_\mathcal{T}$) [42]|%Guaranteed ($\mathcal{C}_\mathcal{W}$) (OURS)|
> |-|:-:|:-:|:-:|
> |PFAUST-M|78 (+-16)|48 (+-18)|82 (+-14)|
> |PFAUST-H|53 (+-16)|30 (+-18)|65 (+-18)|
> ### Several connected components
> Regarding several connected components, it is standard practice for partial shape matching to consider only a single connected component. By definition, consistent pairs can only belong to the same components. Our pipeline would simply generalise by computing the features of each component independently and then concatenating them before estimating the matching, just like other pipelines would.
> ### Complexity
> Regarding the complexity of our method, we would like to point out to the reviewer that we have discussed the computational complexity of the method in appendix B.2. Note that this complexity occurs only during preprocessing, as then our methods does not induce any noticeable additional cost.

---

> > ### Comment · Reviewer_a2Wg · 2024-08-09
> > **Post-Rebuttal**
> >
> > I thank the authors for their clarifications and discussion. I appreciate their insights on CUTS, and I see that for the considered algorithm, it is probably less challenging than HOLES since it should generally preserve more of the geodesic distances of the remaining part.
> >
> > Concerning the Failure cases, I still find the discussion a bit confusing. Since "we do not claim that our regularized approach is always guaranteed to succeed," I would be curious to know more precisely which cases do not succeed.
> >
> > I see the complexity has been analyzed in Appendix B.2; it would also be interesting to know the wall-clock time of an inference. What is the required inference time for evaluating the HOLES test set compared to the DirectMatch baseline?

---

> > > ### Author Response · Authors · 2024-08-10
> > >
> > > We thank the reviewer for the efforts and are happy to have clarified the issue with the CUTS dataset. We here answer all remaining questions.
> > >
> > > ### Failure cases
> > > Failure cases of the regularized approach may happen when relaxing the binary into a soft criterion, as discussed in the rebuttal of reviewer 1VSx, where they claim that indeed this regularization "involves a trade-off between retaining slightly noisy information and maintaining the integrity of the matching process". As such, there might be partial shapes with extremely  complex topologies where this compromise would not be beneficial. Nevertheless, we did not find such failure cases on the data we worked with.
> > >
> > > Note, that as all methods in this field, we do not have a theoretical proof of general optimality. We thus do not claim to have the final word in this fascinating and challenging field. Yet, currently, for the specific task of matching a part to a whole, our empirical evidence indicates that the proposed measure improves current SoTA results by a significant notch.
> > >
> > > ### Inference time
> > > Since the learning architecture we tested is the one proposed in DirectMatchNet, our test-time pipeline is the same, yielding similar inference times. Specifically, inference time is 0.12s for both the proposed method and DirectMatchNet per shape on the HOLES dataset. We will add these run times to the paper.

---

### Official Review · Reviewer_NPnj · 2024-07-11

**Soundness:** 3
**Presentation:** 3
**Contribution:** 2
**Rating:** 6
**Confidence:** 5

**Summary:**

The paper investigates the problem of shape matching in an unsupervised and partial setting. In particular, the paper extends an existing criterion for filtering out potentially inconsistent point pairs. This is done by simply including extrinsic information that bounds from below the distances between pairs of boundary points, resulting in a new criterion that preserves more consistent pairs.

The experiments are conducted for applications including multi-dimensional scaling and partial shape matching, in which the advantage of the proposed loss is demonstrated.

**Strengths:**

The paper is in general well-motivated, and the idea is straightforward to follow. Also, the presentation is clear with sufficient figures to help understanding. the experiments include different applications.

**Weaknesses:**

In general, the paper is well-written, explores a simple yet effective idea, and proves its effectiveness via experiments.

However, the main weakness of this paper comes from the lack of theoretical depth. The proposed method is a modification of an existing idea, whereas new insights are limited. This is also discussed in Limitations of the paper, there are more consistent pairs that could be recovered, yet we have no idea how large the gap is.  That being said, the contribution of the paper is still clear.

**Questions:**

I have no particular critical question. But I am curious, how large the gap is between the preserved consistent pairs and all possibly existing consistent pairs?

**Limitations:**

The limitations have been adequately discussed in the paper.

---

> ### Author Rebuttal · Authors · 2024-08-06
>
> We thank the reviewer for the thoughtful comments and their appreciation of the paper's clear contribution and the soundness and clear presentation of our framework. In the following we relate to the reviewer's remarks and questions.
> ### Theoretical depth
> Regarding the theoretical depth of our paper, we respect the reviewer's perspective. However, we respectfully like to argue otherwise. The proposed criterion, which is indeed a simple deviation from a naive existing one, is less trivial than might seem at first glance. Most existing papers in shape analysis, and in particular in shape matching, focus on using intrinsic properties, such as geodesic distances, LBO eigendecompositions, functional maps, and interactions between non-regular metric spaces. Another philosophy in shape analysis, albeit less popular, is to consider only extrinsic properties, e.g. Wang et al. 2018, but such approaches are ill-suited for problems based on intrinsic properties like shape matching.
>
> In our case, the proposed approach combines intrinsic and extrinsic non-differential quantities. In fact, we invoke an extrinsic measure (embedding distances in a Euclidean space) to analyze intrinsic quantities (geodesic distances). While maybe simple in appearance, such a perspective is far from trivial. In addition, extrinsic Euclidean distances are often ignored in the study of shapes undergoing non-rigid transformations. Finally, our criterion yields theoretically guaranteed pairs (Theorem 2), which is proven in the appendix. Bottom line - it is not a heuristic criterion, but rather a theoretically supported, and apparently practically useful one. Providing a simple approach that integrates intrinsic and extrinsic integral measures, that introduces the often ignored extrinsic Euclidean distances, and is well supported by a solid mathematical theory is a fundamental contribution. Moreover it leads to enhanced empirical performance.
>
> That being said, we do not provide a theoretical guarantee on the measure of consistent pairs our criterion can find, and we are clear on that matter in the limitations paragraph. The same argument is true for other criteria, e.g. $\mathcal{C}\_\mathcal{T}$. Nevertheless, the proposed criterion is theoretically guaranteed to recover all the pairs that the existing criterion $\mathcal{C}\_\mathcal{T}$ finds. We also show in Figure 4 a qualitative difference between the consistent pairs both criteria guarantee.
>
> Y. Wang et al. Steklov spectral geometry for extrinsic shape analysis. ACM TOG, 2018.
> ### Amount of missed consistent pairs
> Regarding the gap between the number of guaranteed pairs via different criteria and the total number of consistent pairs, we conducted experiments during the rebuttal period to estimate it. We evaluated on the datasets PFAUST-H and PFAUST-M the amount of consistent pairs, guaranteed pairs by our criterion $\mathcal{C}\_\mathcal{W}$, and guaranteed pairs by $\mathcal{C}_\mathcal{T}$. We provide in the table below the average percentage of consistent pairs, and among these, the average percentage of guaranteed pairs by each criteria. We also provide the standard deviations in parenthesis.
> ||%Consistent|%Guaranteed ($\mathcal{C}_\mathcal{T}$) [42]|%Guaranteed ($\mathcal{C}_\mathcal{W}$) (OURS)|
> |-|:-:|:-:|:-:|
> |PFAUST-M|78 (+-16)|48 (+-18)|82 (+-14)|
> |PFAUST-H|53 (+-16)|30 (+-18)|65 (+-18)|
>
> We will add this evaluation table to the paper.

---

> > ### Comment · Reviewer_NPnj · 2024-08-13
> >
> > Thanks for the rebuttal! I have no further questions.

---

### Official Review · Reviewer_1VSx · 2024-07-11

**Soundness:** 3
**Presentation:** 3
**Contribution:** 2
**Rating:** 6
**Confidence:** 4

**Summary:**

The paper presents a novel criterion for identifying consistent pairs based on geodesic distances between points on partial and full surfaces. This new loss function utilizes intrinsic geodesic distances, extrinsic distances between boundary points, and a consistency criteria to enhance the performance of partial shape matching. The authors validate the improvements in the loss function both theoretically and through experimental results.

**Strengths:**

- The focus on partial shape matching and unsupervised shape matching for partial to full shapes is intriguing.
- Both quantitative and qualitative evaluations demonstrate significant improvements over the state of the art.

**Weaknesses:**

- The method fully relies on the Euclidean embedding space for boundary distances, which might not always reflect true geodesic distances on the surface, especially in highly non-Euclidean or complex topologies.
-The authors could enhance evaluations by using other correspondence techniques, instead of only using DiffusionNet features, and demonstrate broader generalization capabilities with other signatures.

**Questions:**

- How does the criterion perform on significantly larger datasets with more complex topologies? It would be interesting to integrate a control measure to analyze the effect of filtering inconsistent pairs.

- Regularizing the binary mask into a soft mask M* involves a trade-off between retaining slightly noisy information and maintaining the integrity of the matching process. How is the threshold for the soft mask determined, and how sensitive is the model's performance to this threshold?

- Are there specific scenarios where this regularization might fail?

**Limitations:**

The method might fail in the presence of topological variations, leading to incorrect correspondences if the geodesic paths on the partial surface are substantially different from those on the full surface.

---

> ### Author Rebuttal · Authors · 2024-08-06
>
> We thank the reviewer for the insightful comments and for the appreciation of the strengths of our method when applied to the challenging partial shape matching problem and our significant improvements for this task. We hereby respond to the reviewer's remarks and questions.
> ### Reliance on the Euclidean embedding space
> Regarding the Euclidean term in our criterion, the Euclidean embedding space yields a lower bound on geodesic distances between boundary points. Naturally, this bound is less tight the more the surface is curved. Nevertheless, we demonstrate in both shape matching experiments with highly non-rigid transformations of shapes with complex partial topologies and flattening highly curved embeddings of the Swiss roll, that our criterion is greatly beneficial in non-Euclidean settings. We also ran in the rebuttal period new experiments to evaluate the amount of consistent pairs recovered by our criterion in the PFAUST benchmark and found that we are able to recover most pairs and find about twice as many as the previous criterion which does not rely on Euclidean distances. See the table below, that we will include in the paper, presenting the percentage of consistent pairs, and among those the percentage of guaranteed pairs by the different criteria, along with the standard deviations.
> ||%Consistent|%Guaranteed ($\mathcal{C}_\mathcal{T}$) [42]|%Guaranteed ($\mathcal{C}_\mathcal{W}$) (OURS)|
> |-|:-:|:-:|:-:|
> |PFAUST-M|78 (+-16)|48 (+-18)|82 (+-14)|
> |PFAUST-H|53 (+-16)|30 (+-18)|65 (+-18)|
> ### Generalization
> Regarding the generalization capabilities of our method, we showed the success of our criterion in two completely different tasks: partial shape matching and MDS flattening of partial surfaces. In the learning pipeline, we used the DiffusionNet as it is currently the prominent network for shape correspondence. Nowadays SOTA methods rely on DiffusionNet, and we show that we can improve them even further with our criterion. However, our method is complementary to DiffusionNet, and we could use the proposed criterion to train learnable feature extractors that would potentially replace DiffusionNet in the future.
> ### Larger and more complex datasets
> Regarding the question on the performance of our method on larger datasets with more complex topologies, we evaluated our method on the reference datasets for partial shape matching. We not only evaluated on the leading benchmark in the field SHREC'16, we also tested on the recent benchmark PFAUST to show that our method generalises to other types of shapes with different types of shape partiality. Note that both benchmarks comprise complex topologies due to the way parts were removed. We did not find a larger dataset with more complex topologies for partial shape matching of surfaces undergoing non-rigid transformations.
> ### Control measures
> Regarding control measures for analysing the effect of filtering inconsistent pairs, we point out Table 3 in our ablation study. There, we show how performance changes if we take either a binary or a soft criterion, based either on our wormhole one $\mathcal{C}\_\mathcal{W}$ or $\mathcal{C}\_\mathcal{T}$. In addition, our approach is built on top of DirectMatchNet. Thus, the comparative performance analysis in Tables 1 and 2 shows the effect of including our criterion.
> ### Soft mask matrix and thresholding
> Regarding the soft mask $\boldsymbol{M}^s$, there is actually no threshold parameter in its calculation. It is given directly form the distance matrix $\boldsymbol{D}$ and the $\boldsymbol{K}$ matrix by $\boldsymbol{M}\_{ij}^s = \min(\frac{\boldsymbol{K}\_{ij}}{\boldsymbol{D}\_{ij}},1)$, where $\boldsymbol{K}$ is given by the criterion$$\boldsymbol{K}\_{ij} = \min\limits_{B_1,B_2\in \mathcal{B}} d_{\mathcal{Y}'}(v_i, B_1) + d_{\mathcal{Y}'}(v_j, B_2) + d_E(B_1, B_2),$$where $v_i$ and $v_j$ are the vertices at indices $i$ and $j$ respectively. We will clarify this by adding the definition of $\boldsymbol{K}$. Also, since we want the soft mask matrix $\boldsymbol{M}\_{ij}^s$ to be 1 and equal to the binary mask matrix $\boldsymbol{M}\_{ij}$ for consistent pairs (when $\boldsymbol{D}\_{ij}\le \boldsymbol{K}_{ij}$), we cut off the ratio $\frac{\boldsymbol{K}\_{ij}}{\boldsymbol{D}\_{ij}}$ to $1$ in the definition of the soft mask matrix.
> ### Failure cases
> Regarding failure cases, our binary criterion cannot guarantee inconsistent pairs as proven in Theorem 2. Regarding failure cases due to the regularization, when we would get worse results by switching from the binary to the soft mask, we did not find such scenarios. Nevertheless, we do not claim that our regularized approach would never fail on unseen challenging contexts.

---

> > ### Comment · Reviewer_1VSx · 2024-08-12
> > **Responce to rebuttal**
> >
> > Dear authors,
> >
> > Thank you for submitting your rebuttal and including the additional experiment. I have reviewed the rebuttal and found that my questions have been addressed.
> >
> > 1VSx

---

### Official Review · Reviewer_tftc · 2024-07-19

**Soundness:** 4
**Presentation:** 3
**Contribution:** 3
**Rating:** 7
**Confidence:** 4

**Summary:**

This paper introduces a refined criterion for detecting pairs of points whose geodesic distance on a partial surface is equal to that on the full surface. The "wormhole" criterion is sound and less conservative than previous such criteria. The refined criterion improves results in planar embedding with multidimensional scaling (MDS) as well as partial-to-whole shape matching.

**Strengths:**

This paper is clear and well-written. It is focused on one core contribution, the wormhole criterion for guaranteed pairs of points. And the results on both MDS and partial matching validate the utility of the refined criterion.

**Weaknesses:**

### Miscellaneous
- This is confusing: "This condition naturally follows from the fact that the shortest path in the full surface $\mathcal{Y}$ is shorter than any path between the points passing through the boundary $\mathcal{B}$, and in particular, ones that pass through the closest boundary points" (137-139). I think what you mean is either that (1) in general, the length of a path between the points in the full surface that passes through the boundary is at least as long as the sum of their distances to the boundary; or that (2) for pairs of points that satisfy the condition $\mathcal{C}_{\mathcal{T}}$, the shortest path in the full surface is *no shorter* than the path in the partial surface.
- $\mathbf{K}$ is only mentioned in passing on line 170 before being used in equation (6). It might be helpful to introduce more explicitly what you mean by the "threshold" matrix, though I think it is inferable from context.
- Line 21: "Consequentially" should be "Consequently"

**Questions:**

- You say you could "easily generalize the discussion" to the case of both surfaces being partial (103). Would this just amount to using the intersection of the criteria on both sides?
- The boundary to which distances are measured is viewed as the boundary of $\mathcal{Y}'$ excluding the boundary of $\mathcal{Y}$. Given a partial surface, how do you know which parts of its boundary are part of the boundary of the full surface?
- Why is the loss function scaled by the vertex area twice in equation (5)? It might be helpful to state this as an integral in the smooth setting before discretizing.

---

> ### Author Rebuttal · Authors · 2024-08-06
>
> We thank the reviewer for the detailed comments and for recognizing the utility of the wormhole criterion in both MDS and partial-to-whole shape matching tasks. We are happy that the reviewer enjoyed the clarity, soundness, and contribution in our manuscript. We fixed the typos pointed out by the reviewer and added the definition of $\boldsymbol{K}$ to the paper, $\boldsymbol{K}\_{ij} = \min\limits_{B_1,B_2\in \mathcal{B}} d_{\mathcal{Y}'}(v_i, B_1) + d_{\mathcal{Y}'}(v_j, B_2) + d_E(B_1, B_2)$, where $v_i$ and $v_j$ are $i$-th and $j$-th vertices. In the following we respond to the reviewer's remarks and questions.
> ### Line 137
> Regarding the confusion in lines (137-139), we will rewrite this sentence to avoid confusion by taking a formulation similar to version (1) suggested by the reviewer.
> ### When both shapes are partial
> Regarding our theoretical discussion about consistent pairs, we focused on the case when only one surface is partial. In our experiments, we then used the found consistent pairs to guide the learning process for partial-to-whole shape matching. If both shapes are partial, the theory allows to find consistent pairs on each partial surface independently. In practice though, we would need to find the shared consistent pairs between both partial surfaces should we want to learn the matching between both partial surfaces. This should indeed be done by taking some form of intersection of found consistent pairs. Designing a partial-to-partial shape matching pipeline using the proposed criterion is left to future work (we will emphasize it in Limitations section).
> ### Boundaries
> Regarding boundaries, in practice, we treat all boundaries equally. We will remove Line 133 "excluding the boundaries of $\mathcal{Y}$" and clarify in the paper that all boundaries are treated equally.
> ### Double area scaling
> Regarding the double area scaling in Equation (5), our loss function is based on the continuous loss presented by Aflalo et al. IJCV 2016, see also [11]. Following their formulations we introduce a loss for the correspondence function between $\mathcal{X}$ and $\mathcal{Y}'$, defined by $\tilde{p}:\mathcal{Y'}\times \mathcal{X} \rightarrow \mathbb{R}^+$ as,
> $$\int_\mathcal{Y'\times Y'} \left(\int_\mathcal{X\times X} d_\mathcal{X}(x_1, x_2) \tilde{p}(x_1, y'\_1)\tilde{p}(x_2, y'\_2) da_{x_1} da_{x_2} - d_\mathcal{Y'}(y'\_1, y'\_2)\right)^2 m(y'\_1,y'\_2) da_{y'\_1} da_{y'\_2} $$where $d_{\mathcal{X}}$ and $d_{\mathcal{Y}'}$ measure the distances between surface points, and $m:\mathcal{Y'}\times \mathcal{Y'} \rightarrow \\{0,1\\}$ is our binary masking function. We readily have the discrete version,$$\mathcal{L}(\boldsymbol{\tilde P}, \boldsymbol{D}\_\mathcal{X}, \boldsymbol{D}\_\mathcal{Y'},\boldsymbol{M})=\||\boldsymbol{M}\odot(\boldsymbol{\tilde P} \boldsymbol{A}\_\mathcal{X} \boldsymbol{D}\_\mathcal{X}\boldsymbol{A}\_\mathcal{X}\boldsymbol{\tilde P}^\top - \boldsymbol{D}\_\mathcal{Y} )\||\_\mathcal{Y'Y'},$$ where $\||\boldsymbol{B}\||\_\mathcal{Y'Y'} = \text{trace}(\boldsymbol{B}^\top\boldsymbol{A}\_\mathcal{Y}\boldsymbol{B}\boldsymbol{A}\_\mathcal{Y})$. Similar to previous papers [21] we assume that our learned correspondence matrix implicitly contains the area matrix $\boldsymbol{A}\_\mathcal{X}$, thus, $\boldsymbol{P} = \boldsymbol{\tilde P} \boldsymbol{A}\_\mathcal{X}$. Consequently, our loss function is defined as follows,$$\mathcal{L}\_{\text{geo}}(\boldsymbol{P}, \boldsymbol{D}\_{\mathcal{X}}, \boldsymbol{D}\_{\mathcal{Y}'}, \boldsymbol{M})=\||\boldsymbol{M}\odot(\boldsymbol{PD}\_\mathcal{X}\boldsymbol{P}^\top - \boldsymbol{D}\_\mathcal{Y'}) \||\_\mathcal{Y'Y'}.$$For simplicity, we expressed it element-wise,$$\mathcal{L}\_{\text{geo}}(\boldsymbol{P}, \boldsymbol{D}\_{\mathcal{X}}, \boldsymbol{D}\_{\mathcal{Y}'}, \boldsymbol{M})=\sum_{ij} \boldsymbol{M}\_{ij} \boldsymbol{A}\_{\mathcal{Y}'ii} \boldsymbol{A}\_{\mathcal{Y}'jj} \left( (\boldsymbol{P}\boldsymbol{D}\_\mathcal{X}\boldsymbol{P}^\top - \boldsymbol{D}\_{\mathcal{Y}'})_{ij} \right)^2.$$We will add a derivation of the loss from its continuous to the discrete versions.
>
> Y. Aflalo et al. Spectral generalized multidimensional scaling. IJCV, 2016

---

> > ### Comment · Reviewer_tftc · 2024-08-14
> >
> > Thank you to the authors for their detailed response. I continue to think that the wormhole criterion will be of use to the community.

---

### Decision · Program_Chairs · 2024-09-25

**Decision:**

Accept (poster)

**Comment:**

All four reviewers unanimously recommend acceptance after the rebuttal. The paper is well-written and there is a clear contribution which has been demonstrated to work well in practice. The AC concurs and recommends accept as well.